# FROM SEEING TO EXPERIENCING: SCALING NAVIGATION FOUNDATION MODELS WITH REINFORCEMENT LEARNING

**Honglin He**[1,*], **Yukai Ma**[1,*], **Brad Squicciarini**[2], **Wayne Wu**[1], **Bolei Zhou**[1]
[1]University of California, Los Angeles
[2]Coco Robotics
https://vail-ucla.github.io/S2E

## ABSTRACT

Navigation foundation models trained on massive web-scale data enable agents to generalize across diverse environments and embodiments. However, these models, which are trained solely on offline data, often lack the capacity to reason about the consequences of their actions or adapt through counterfactual understanding. They thus face significant limitations in the real-world urban navigation where interactive and safe behaviors, such as avoiding obstacles and moving pedestrians, are critical. To tackle these challenges, we introduce the Seeing-to-Experiencing (S2E) learning framework to scale the capability of navigation foundation models with reinforcement learning. S2E combines the strengths of pre-training on offline videos and post-training through reinforcement learning. It maintains the model's generalizability acquired from large-scale real-world videos while enhancing its interactivity through reinforcement learning in simulation environments. Specifically, we introduce two technical innovations: 1) an Anchor-Guided Distribution Matching strategy for offline pretraining, which stabilizes learning and models diverse motion patterns through anchor-based supervision; and 2) a Residual-Attention Module for reinforcement learning, which obtains reactive behaviors from simulation environments without erasing the model's pretrained knowledge. Moreover, we establish a comprehensive end-to-end evaluation benchmark, NavBench-GS, built on photorealistic 3D Gaussian Splatting reconstructions of real-world scenes that incorporate physical interactions. It can systematically assess the generalizability and safety of navigation models. Extensive experiments show that S2E mitigates the diminishing returns often seen when scaling with offline data alone. We perform a thorough analysis of the benefits of Reinforcement Learning (RL) compared to Supervised Fine-Tuning (SFT) in the context of post-training for robot learning. Our findings emphasize the crucial role of integrating interactive online experiences to scale foundation models in Robotics. Code will be made available.

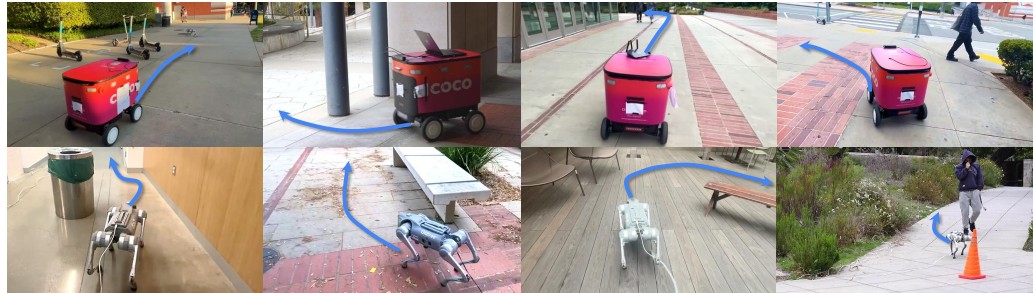

Real-world deployments of S2E on wheeled and quadruped robots

Figure 1: **Real-world deployments of S2E.** S2E achieves zero-shot generalization across environments and embodiments. We demonstrate its effectiveness on wheeled and quadruped robots in diverse urban scenarios.

---

*  Equal contribution.

# 1 INTRODUCTION

Foundation models have demonstrated transformative capabilities across various domains, including language understanding (Touvron et al., 2023; Bai et al., 2023), generation (Rombach et al., 2021; Lin et al., 2024), and visual recognition (Wang et al., 2023; Kirillov et al., 2023; Yang et al., 2024). Through training on massive data, these models significantly enhance the generalizability and adaptability of downstream tasks. However, applying foundation models to robot navigation presents unique challenges (Firoozi et al., 2023) due to the complex nature of sequential decision-making in dynamic real-world environments. For example, in a bustling urban space, navigation foundation models must make real-time decisions to avoid collisions with obstacles, such as trash bins, and safely maneuver through ever-changing crowds.

Recent work on navigation foundation models has harnessed large-scale web videos and human demonstrations for pretraining (Shah et al., 2023a;b; Sridhar et al., 2024). These approaches primarily rely on passive visual learning (Kim et al., 2024; Brohan et al., 2022), where models are trained to imitate behaviors in massive video data. Such data captures diverse visual observations of the real world; yet, it lacks explicit information on physics and cause-and-effect relations, which are crucial for decision-making. As a result, navigation policies trained solely on offline data often exhibit limited *reactivity* to the surroundings and struggle to adapt to diverse objects and motions.

While *offline* video data helps build proper perceptual prior for the model, it only captures statistical correlations, not grounded causality (Silver & Sutton, 2025). Visual imitation (Dai et al., 2025; Ren et al., 2025) teaches an agent what actions look like, but *not* how to adapt, recover, or reason about counterfactual outcomes when the environment changes. Thus, navigation foundation models must move **from seeing to experiencing**: *actively* interact with the world, receive feedback, and refine behaviors through trial and error. As shown in Figure 2, similar to humans learning skateboarding, where experience with balance, falling, and correction is irreplaceable, agents must interact with the world to acquire true adaptability. Reinforcement Learning (RL) enables agents to bridge the gap between observations and actions, offering a scalable interactive learning paradigm that enriches model capabilities beyond the behavior cloning of static datasets. However, RL alone has shown limited success in building generalizable navigation models. Prior approaches (Shen et al., 2019; Putta et al., 2024; Truong et al., 2021; Lin & Yu, 2025; Xie et al., 2025) have trained agents in narrow synthetic environments using RL; however, due to poor sampling efficiency and the lack of inductive priors, models struggle to achieve scalable and generalizable navigation capabilities in the real world.

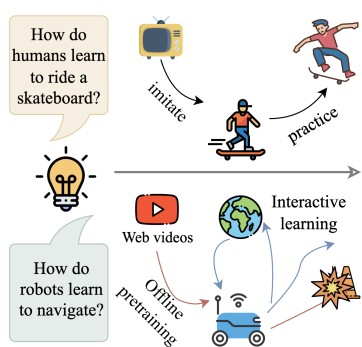

Figure 2: **Motivation.** Like humans, AI agents must also go through interactive practices and learn from feedback to obtain actionable skills.

To this end, we propose a new learning framework, **S**eeing-**to**-**E**xperiencing (**S2E**), to scale navigation foundation models with reinforcement learning in simulated environments while maintaining the generalizable visual representation acquired through pretraining on offline videos. This framework comprises two crucial technical components. First, we introduce Anchor-Guided Distribution Matching (AGDM), a strategy for pre-training on real-world video data. It is designed as an anchor-guided model architecture to learn complex multimodal distributions in normalized motion trajectory space, thereby enabling the efficient representation of diverse behaviors across various scenarios. This design significantly mitigates learning uncertainty and provides a more reliable backbone to ease subsequent online adaptation. Moreover, the anchor-based architecture naturally supports the cross-embodiment deployment of the foundation model. Second, we propose a Residual-Attention Module (RAM) for RL post-training in simulation environments. It is designed as a residual architecture by copying the pretrained attention block and learning a residual component that selectively captures knowledge acquired from online interactions. This design enables agents to acquire novel capabilities, such as obstacle avoidance and movement anticipation, through reinforcement learning while preserving the generalizable visuomotor representations learned from offline pre-training.

We further introduce NavBench-GS, a comprehensive end-to-end evaluation benchmark built on realistic 3D Gaussian Splatting environments with accurate physics and interactive dynamics. Unlike

prior evaluations that rely on offline 2D video-based testing (Shah et al., 2023a;b; Sridhar et al., 2024), NavBench-GS enables reactive simulation to facilitate closed-loop policy assessment in photo-realistic 3D scenes. These scenes, combining realistic visual appearance and physical interaction with reproducibility, address a longstanding challenge in robotics: the difficulty of replicating real-world environments for end-to-end evaluation. It enables standardized, reproducible evaluation of navigation foundation models in terms of generalization and safety in unseen settings.

Extensive experiments show that reinforcement learning substantially enhances the policy performance and alleviates the diminishing returns associated with scaling solely on offline data. We analyze the effectiveness of Reinforcement Learning (RL) versus Supervised Fine-Tuning (SFT) in post-training for robot learning. Although both methods are widely discussed in relation to large language models (LLMs) (Kumar et al., 2025), they remain underexplored in the field of scaling robot learning. Additionally, we demonstrate the generalizability of the proposed S2E framework through real-world evaluations in challenging scenarios.

## 2 RELATED WORK

**Goal conditioned navigation.** Goal conditioned navigation is the most common setting for robotic navigation. Prior works have developed diverse approaches to represent navigation goals, which can be categorized into three main paradigms: 1) image-goal navigation, where target images serve as visual references (Mezghan et al., 2022; Ramakrishnan et al., 2022; Zhu et al., 2017); 2) position-goal navigation, which directly encodes destination coordinates (Chaplot et al., 2020a; Chattopadhyay et al., 2021; Gordon et al., 2019); and 3) object-goal navigation, where targets are specified through object categories (Al-Halah et al., 2022; Chang et al., 2020; Chaplot et al., 2020b).

Deep reinforcement learning (Mirowski et al., 2016) has demonstrated promising results in mapless navigation by eliminating dependency on maps. However, these methods often suffer from limited generalization, particularly in visual navigation (Shen et al., 2019; Putta et al., 2024; Truong et al., 2021), due to the constrained diversity of training scenarios. The synthetic nature and restricted variation in simulated training worlds inherently limit the policy's ability to adapt to real-world complexity and unseen scenarios.

**Navigation foundation models.** Many recent works have proposed various vision-based navigation foundation models (Shah et al., 2023a;b; Sridhar et al., 2024), leveraging advantages in cross-sensor capabilities and rich vision data for improved generalizability across different robot platforms and camera configurations. CityWalker (Liu et al., 2024b) and NWM (Bar et al., 2024) further enhance the environmental comprehension of policy by incorporating future state prediction, enabling more informed navigation decisions. However, a key limitation of such approaches is the lack of environmental interactions in the training data, which, as we demonstrate in Section 4.2, results in poor performance in obstacle and pedestrian avoidance. To address this, it is essential to develop policies that are generalizable and capable of high-quality local planning, rather than relying solely on path-following capabilities.

**Hybrid learning with pretraining and finetuning.** The paradigm of offline pretraining followed by RL fine-tuning has emerged as a powerful framework for training robust control policies, bridging the gap between data efficiency and real-world adaptability. Early successes in playing games, such as AlphaGo (Silver et al., 2016) and AlphaStar (Arulkumaran et al., 2019), demonstrated the potential of combining large-scale pretraining (*e.g.*, supervised learning from expert trajectories) with RL fine-tuning to achieve superhuman performance. This approach has since been extended to train foundation models, where pre-trained LLMs (Touvron et al., 2023; Halterman & Keith, 2025) and VLMs (Chen et al., 2024) are fine-tuned via Reinforcement Learning from Human Feedback (RLHF) to align with human values and preferences.

## 3 S2E LEANING FRAMEWORK

To address the challenge of training generalizable and interactive foundation navigation models, we propose **S2E**, a hybrid learning framework that combines pretraining on videos and reinforcement learning in simulated environments. Figure 3 illustrates the overall framework of the proposed S2E.

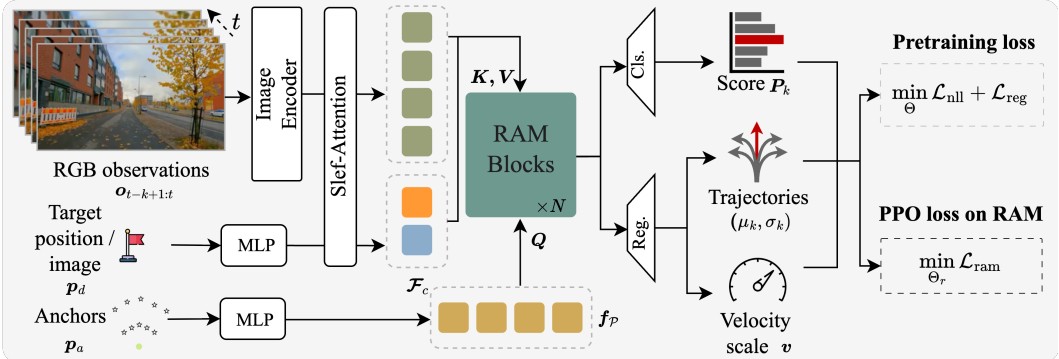

Figure 3: **Illustration of S2E framework.** The model receives continuous RGB frames as context information, goal point or goal image as guidance, and uses spatial anchors as queries for prediction. First, context embeddings are fused via a self-attention module. The outputs are then used as keys ($K$) and values ($V$). Meanwhile, the anchor features $\boldsymbol{f_P}$ serve as queries ($Q$). Subsequently, RAM blocks compute weighted features from $K$ and $V$ based on the anchor queries $Q$, and produce refined anchor features. A classification and a regression head decode the anchor features to predict scores and normalized trajectories with a velocity scale. In the pretraining stage, the model is trained end-to-end with NLL and regression loss (Equation 2). In the fine-tuning stage, only the parameters within the RAM blocks are optimized using the policy gradient from $\mathcal{L}_{\text{ram}}$.

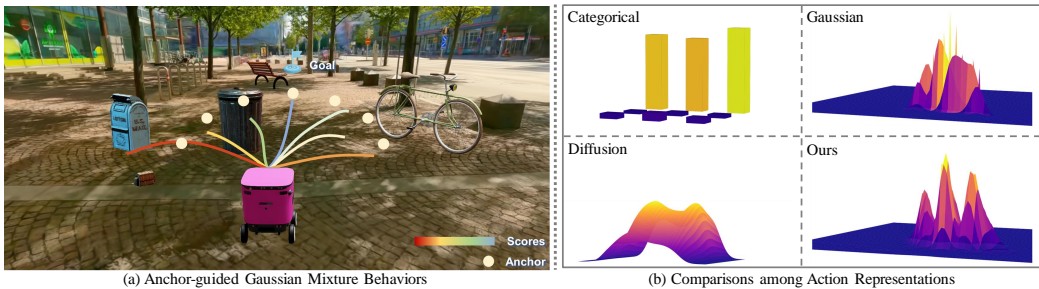

(a) Anchor-guided Gaussian Mixture Behaviors  (b) Comparisons among Action Representations

Figure 4: **Illustration of anchor-guided distribution matching.** (a) Illustration of anchor-guided Gaussian Mixture behaviors in a sidewalk scenario, where anchors guide diverse behavior generation. (b) Comparison of predicted action distributions between different representations.

It aims to learn a visual navigation policy $\pi$ that enables a robot to navigate from the source waypoint coordinate $\boldsymbol{p}_s$ to the target waypoint coordinate $\boldsymbol{p}_d$. This task focuses on local navigation (Zhu et al., 2017; Liu et al., 2024b) between consecutive waypoints, which can be easily extended to long-distance tasks by chaining waypoints obtained from path planning (Thrun, 2002) or GPS. At each timestep $t$, the observations are RGB frames $\boldsymbol{o}_{t-k+1:t}$ spanning the past $k$ frames, and the target coordinates $\boldsymbol{p}_d$ or target image $\boldsymbol{I}_d$. The output is a short-term relative waypoint as its action $\boldsymbol{a}$, then locomotion models can execute it to move the robot incrementally toward the goal. Our framework consists of two key technical components: 1) Anchor-guided distribution matching for pretraining generalizable backbone representation (Section 3.1), and 2) Residual-Attention Module for injecting obstacle avoidance capability with RL (Section 3.2).

## 3.1 PRETRAINING WITH ANCHOR-GUIDED DISTRIBUTION MATCHING

A backbone for extracting meaningful visual features is essential for generalization across diverse scenarios. We pretrain our model on 100 hours of navigation videos collected from various robots and platforms (Shah et al., 2021; Karnan et al., 2022b;a; Liu et al., 2024a; 2025; Hirose et al., 2025), covering a wide range of environments. However, modeling *multi-modal data distributions* still presents significant challenges (Sridhar et al., 2024), as the model must generate diverse predictions under the same conditions. To address this, we introduce the anchor-guided distribution matching.

**Anchor-guided distribution matching.** Robot navigation trajectories are inherently multimodal – given the same observation, multiple distinct actions may be valid. Effectively modeling this multimodality is crucial for generalizable policies. However, common representations like discrete actions (Shafiullah et al., 2022) or unimodal Gaussians (Shah et al., 2023a;b; Liu et al., 2024b)

lack expressiveness, limiting the model to capture the information from partial observation, the accumulation of prediction errors over time (Codevilla et al., 2019), etc. While diffusion models (Chi et al., 2023; Sridhar et al., 2024), though expressive, are overly flexible and difficult to control in navigation, often yielding fragmented trajectories that compromise safety and stability.

To address this, we propose an anchor-guided Gaussian Mixture Model (GMM) (Gu et al., 2021; Shi et al., 2022) to represent robot actions in urban navigation, as illustrated in Figure 4 (a). This formulation offers both multimodality and structure. Anchors, uniformly sampled in the robot's forward direction, serve as interpretable high-level intentions. Each anchor corresponds to a Gaussian mode in the mixture, with learned scores reflecting likelihoods. The policy learns to generate and choose trajectories based on these anchors, enabling diverse yet goal-aligned behaviors. This approach combines expressiveness with training stability and is also well-suited for fine-tuning with RL.

We present the distributions of actions using various representations in Figure 4 (b), including categorical, unimodal Gaussians, diffusion policy, and our anchor-guided Gaussian Mixture Model (GMM). The categorical representation learns actions in discretized bins, which restricts its ability to capture nuanced or multimodal behavior. The unimodal Gaussian representation tends towards a single modal distribution, making it ineffective in capturing the multiple valid actions in trajectories. The diffusion policy learns a smooth and comprehensive distribution, but is too flexible, which complicates the guidance and control of behaviors. Our anchor-guided GMM provides a structured, multi-modal distribution, where different anchors specialize in different high-level intentions (*e.g.*, going straight, turning, yielding). Also, the model retains intra-mode uncertainty by allowing moderate variance within each anchor's predicted distribution with a learned standard deviation.

The model architecture for distribution matching is illustrated in Figure 3. Specifically, we generate $M$ representative intention points $\boldsymbol{p}_a \in \mathbb{R}^{M \times 2}$ by K-Means (Lloyd, 1982) on the unified dataset, which serves as an additional input beyond RGB frames and target position. The distribution of the action $w_t$ at timestep $t$ under observation $\boldsymbol{o}_{t-k+1:t}$ is a Gaussian Mixture Model (GMM) defined as:

$$\boldsymbol{q}(w_t|\boldsymbol{o}_{t-k+1:t}) = \sum_{m=1}^{M} q_m \cdot \mathcal{N}_m(w_x - \mu_x^m, \sigma_x^m;\ w_y - \mu_y^m, \sigma_y^m; \rho^m), \tag{1}$$

where the score distribution $q_m$ denotes the probability of each intention point $\boldsymbol{p}_a^m$ being selected, $\boldsymbol{w}_t = (w_x, w_y)$ denotes the position as input of the locomotion to generate action $a$. $\mu_{x/y}^m$, $\sigma_{x/y}^m$, and $\rho^m$ denote the mean, standard deviation, and correlation predicted by the $m$-th trajectory head, respectively. Additionally, we predict a scale $v \in R^+$ per mode, allowing the policy to model absolute trajectory magnitude while preserving directionality.

**Training objective.** The model is trained end-to-end with two dense training losses $L_{nll,i}$ and $L_{reg,i}$ on each prediction head after each RAM block $i$. The first is the Negative Log-Likelihood (NLL) loss as shown in Equation 2 used to supervise both the classification and trajectory heads. Inspired by Shi et al. (2022), we employ an assignment strategy that selects the mode whose predicted direction best aligns with the ground-truth trajectory for optimization. The second loss is an L2 regression loss used to optimize the velocity scale,

$$\mathcal{L}_{nll,i} = -\log \mathcal{N}_h(\hat{w}_x - \mu_x^h, \sigma_x^h; \hat{w}_y - \mu_y^h, \sigma_y^h; \rho^h) - \log(q_h), \tag{2}$$

$$\mathcal{L}_{reg,i} = ||\hat{v} - v||_2^2, \tag{3}$$

where the selected mode $h$ is the one whose anchor is closest to the ground truth and chosen for optimization. More details about pretraining on video data are provided in the App. E.2 and App. E.3.

## 3.2 Reinforcement Learning with Residual Attention Module

To enhance the specific interaction capabilities of a pre-trained navigation model, a closed-loop RL phase is essential. While imitation learning provides a strong initialization, it inherently lacks the mechanism to correct *covariate shift*, *i.e.*, the divergence between the training distribution and induced trajectory from the policy. When encountering out-of-distribution (OOD) states, prior from offline data become unreliable. RL addresses this failure mode by providing online, corrective feedback, allowing the policy to learn inductive biases for recovery and fine-grained manipulation that are absent in static datasets. To achieve this goal, a naive approach is to fine-tune *all* parameters using RL. However, such strategies introduce two fundamental problems.

*Forgetting of pre-trained capabilities (FPC).* Previous works (Wołczyk et al., 2024; Schmied et al., 2023) show that full-parameter fine-tuning in RL can cause a pretrained policy to lose behaviors it previously performed well. This degradation arises from interference in the function approximator during adaptation and is particularly problematic in transfer RL. When the distribution of visited states shifts, the pretrained capabilities in the under-visited regions deteriorate significantly.

*Domain shift.* Full-parameter RL fine-tuning also exposes the model to a severe observation level domain shift. Let $D_{\text{real}}(\boldsymbol{o})$ denote the real-world observation distribution and $D_{\text{sim}}(\boldsymbol{o})$ the simulator observation distribution. These distributions differ significantly in texture, lighting, object appearance, and sensor noise, i.e., $D_{\text{real}}(\boldsymbol{o}) \neq D_{\text{sim}}(\boldsymbol{o})$. When parameters of the observation encoder $\Theta_{\text{E}}$ are updated only using observations $o \sim D_{\text{sim}}(\boldsymbol{o})$, the learned feature extractor $\mathcal{F}_{\Theta_{\text{E}}}^{\text{sim}}$ is optimized on the simulator domain $D_{\text{sim}}(\boldsymbol{o})$. However, when deployment, the policy receives images from $D_{\text{real}}(\boldsymbol{o})$. There would be a feature difference between the feature from the pre-trained feature extractor $\mathcal{F}_{\Theta_{\text{E}}}^{\text{pre}}$ and the fine-tuned one $\mathcal{F}_{\Theta_{\text{E}}}^{\text{sim}}$, measured by

$$\Delta_{\text{feat}} = \left\| \mathbb{E}_{o \sim D_{\text{real}}} \left[ \mathcal{F}_{\Theta_{\text{E}}}^{\text{sim}}(\boldsymbol{o}) \right] - \mathbb{E}_{o \sim D_{\text{real}}} \left[ \mathcal{F}_{\Theta_{\text{E}}}^{\text{pre}}(\boldsymbol{o}) \right] \right\|. \tag{4}$$

Since $D_{\text{sim}}$ and $D_{\text{real}}$ differ at the pixel level, full-model RL fine-tuning yields $\|\Delta_{\text{feat}}\|$ that grows rapidly, reflecting that the encoder overfits to simulated RGB statistics and no longer produces the pretrained representation, significantly degrading real-world performance.

**Residual attention module.** To address these challenges, a more selective form of fine-tuning is required; that is, we fine-tune only the modules that can enhance task-specific adaptation while avoiding interference with the pretrained visual representations and preventing degradation of previously acquired capabilities. In navigation task, we aim to fine-tune components that are tightly coupled with agent-environment interaction yet robust to sim-to-real gaps. We identify *cross-attention* layers as the ideal target. Unlike visual encoders $\phi_V$ or self-attention layers, which process raw scene textures and are highly sensitive to domain shifts (Tobin et al., 2017; Sridhar et al., 2024), cross-attention explicitly models the relationship between the agent and the environment. The architectural role of cross-attention is computing

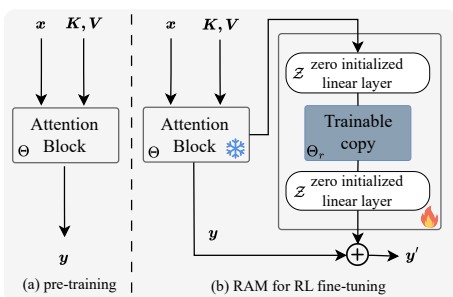

Figure 5: **Residual attention module.**

$$\text{Attn}(\boldsymbol{Q}, \boldsymbol{K}, \boldsymbol{V}) = \text{softmax}\left( \frac{\boldsymbol{Q}\boldsymbol{K}^{\top}}{\sqrt{d}} \right) \boldsymbol{V}, \tag{5}$$

where $\boldsymbol{Q}$ encodes the agent state (e.g., trajectory tokens) and $(\boldsymbol{K}, \boldsymbol{V})$ encodes observation features. This operation explicitly binds agent behavior to environmental context and primarily captures *relational* structure, which is far more stable under appearance changes than raw visual features.

To fine-tune this mechanism effectively, we introduce the **R**esidual-**A**ttention **M**odule (**RAM**), as illustrated in Figure 5. Inspired by previous works (Zhang et al., 2023; Wu et al., 2024; Alayrac et al., 2022; Li et al., 2023), we freeze the original pre-trained parameters $\Theta_D$ of the cross-attention layer $\psi_D$ to preserve generalized capabilities and introduce a parallel, trainable copy $\Theta_l$. This copy is gated by zero-initialized linear layers $\mathcal{Z}$. Formally, the adapted output $\boldsymbol{Q}'$ is computed as

$$\boldsymbol{Q}' = \psi_D(\boldsymbol{Q}; \boldsymbol{K}, \boldsymbol{V}; \Theta_D) + \mathcal{Z}\left( \psi_D(\mathcal{Z}(\boldsymbol{Q}); \boldsymbol{K}, \boldsymbol{V}; \Theta_l) \right). \tag{6}$$

The zero-initialization of $\mathcal{Z}$ guaranties that at the start of fine-tuning, the contribution of the residual branch is null, ensuring $\boldsymbol{Q}' = \psi_D(\boldsymbol{Q}; \boldsymbol{K}, \boldsymbol{V}; \Theta_D)$. This mechanism creates a structural curriculum via the gradient flow. Since $\mathcal{Z}$ is a linear projection initialized to zero, the backpropagated gradient to the adapter parameters,

$$\nabla_{\Theta_l} \mathcal{L} \propto \frac{\partial \mathcal{L}}{\partial \mathcal{Z}} \cdot W_{\mathcal{Z}}, \tag{7}$$

initially vanishes. Consequently, the adaptation branch remains dormant during the high-variance phase of early RL exploration and only becomes active as the gate weights $W_{\mathcal{Z}}$ gradually move from zero, allowing for a controlled, progressive injection of interaction dynamics.

As an additional advantage, this design is substantially more parameter- and computation-efficient than full-parameter fine-tuning. In practice, the full-sized model with parameters $\Theta_0$ (Hu et al., 2022) contains millions to billions of parameters, learning a full-sized update $\Delta\Theta$ from the simulator is computationally expensive. Each iteration requires full forward–backward passes through the entire encoder and decoder, implemented by many transformer blocks, dramatically increasing memory usage. With our approach, *i.e.*, instead of updating the full model, we train a lightweight plugin module whose parameter $\Theta_l$ satisfies $|\Delta\Theta_l| \ll |\Theta_0|$, while omitting costly updates to the vision encoder (and thereby avoiding the domain-shift issue) and still improving the policy performance.

**Reward function design.** In reinforcement learning, the reward function provides the fundamental learning signal that shapes agent behaviors by reinforcing desirable outcomes and penalizing undesirable ones. We design the reward progressively, moving from essential objectives to higher-level refinements, $R = R_{\mathcal{G}} + R_{\mathcal{R}} + R_{\mathcal{H}}$, where

- Global goal $R_{\mathcal{G}}$ encourages the agent to efficiently reach the destination while ensuring basic safety. To achieve this goal, we employ four terms—dense/sparse goal-reaching $R_{g,d}, R_{g,s}$ and dense/sparse collision penalties $R_{c,d}, R_{c,s}$ together.
- Rule regularization $R_{\mathcal{R}}$ enforces general navigation rules, such as sidewalk centering and social compliance.
- Human likeness $R_{\mathcal{H}}$ encourages smooth, natural, and interpretable behaviors that align with human navigation patterns.

The details on each reward item can be found in App. E.5.

**Training objective.** The pretrained model outputs a waypoint trajectory $w_t \in \mathbb{R}^{10\times2}$, which is robot-agnostic. To enable training with dynamics-aware robots in the simulator, we employ a differentiable controller $\mathcal{F}_d$ that takes $w_t$ as input and generates velocity commands for the locomotion module $\mathcal{F}_l$. We only finetune the parameters $\Theta_r$ of additional branches from RAM blocks, so the gradients from the context features $V_t = \text{Encoder}(o_{t-k+1:t})$ and $f_{\mathcal{P}}$ are truncated. To optimize the policy, we employ a PPO-based objective with entropy regularization:

$$\min_{\Theta_r} \mathcal{L}_{\text{ram}} = -\mathcal{L}_{\text{policy}} + \alpha\mathcal{L}_{\text{value}} - \beta\,\mathcal{H}_\pi, \tag{8}$$

$$\mathcal{L}_{\text{policy}} = \mathbb{E}_t[\min(r_t\hat{A}_t,\ \text{clip}(r_t, 1-\epsilon, 1+\epsilon)\hat{A}_t)], \tag{9}$$

$$\mathcal{L}_{\text{value}} = \mathbb{E}_t\left[\left(V_\phi(o_{t-k+1:t}) - R_t\right)^2\right], \tag{10}$$

$$\mathcal{H}_\pi \approx \sum_{m=1}^{M} q_m \cdot [\frac{1}{2}\log[(2\pi e)^2\sigma_x^{m2}\sigma_y^{m2}]] - \sum_{m=1}^{M} q_m \log q_m, \tag{11}$$

where $r_t = \frac{\pi(a_t|s_t)}{\pi_{\text{old}}(a_t|s_t)}$ is the probability ratio, $\hat{A}_t$ is the estimated advantage, and $\epsilon$ is the clipping threshold, $V_\phi$ is the value network that takes the context feature as input and use an MLP to output the value prediction. $\mathcal{H}_\pi$ is a simplified approximation of the entropy of the GMM that does not admit a closed-form analytic solution for the KL divergence. Additional details are provided in the App. E.4.

## 4 EXPERIMENTS

In this section, we provide our experimental results. Section 4.1 validates the effectiveness of RL in further scaling navigation foundation models. Section 4.2 benchmarks state-of-the-art navigation foundation models in realistic 3DGS scenes. Section 4.3 presents real-world evaluation results and demonstrates the generalizability of S2E. For more details, please refer to the App. C and App. D.

### 4.1 SCALING UP MODEL PERFORMANCE VIA REINFORCEMENT LEARNING

We first validate our motivation: while large-scale offline pre-training yields broad generalization, we investigate if RL can improve the performance after pretraining by enabling the model to learn

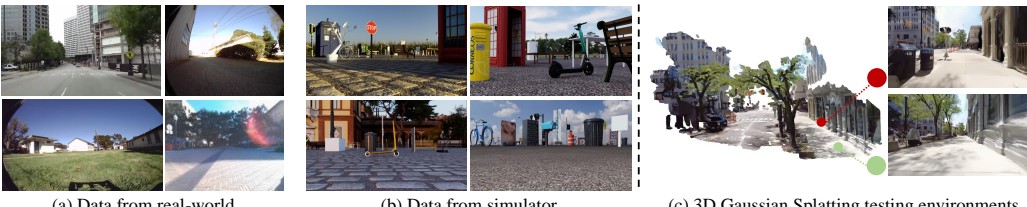

| (a) Data from real-world | (b) Data from simulator | (c) 3D Gaussian Splatting testing environments |

Figure 6: **Overview of training environments and evaluation benchmark.** (a) Real-world data has realistic appearances but lacks interactions. (b) Synthetic data from simulator supplements rich physical interactions. (c) Scene in NavBench-GS offers realistic visual appearances and physical interactions for E2E evaluation.

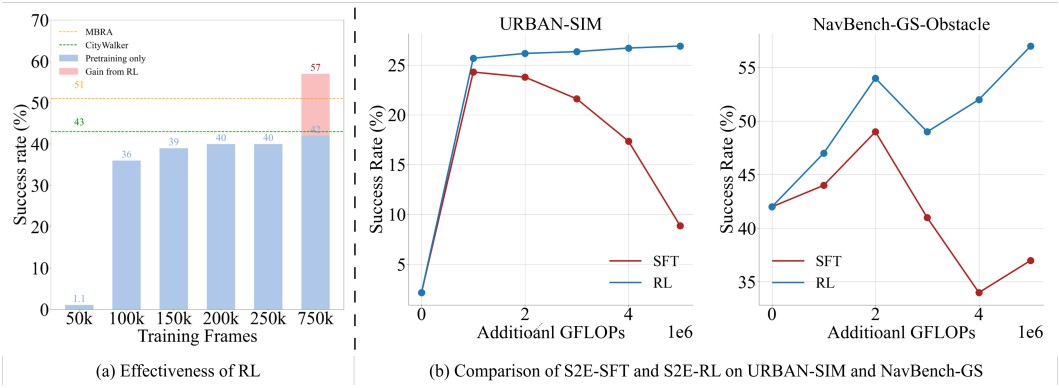

(a) Effectiveness of RL      (b) Comparison of S2E-SFT and S2E-RL on URBAN-SIM and NavBench-GS

Figure 7: **Effectiveness of reinforcement learning.** (a) Success rates of policies trained with varying amounts of data, showing gain from RL fine-tuning over only supervised learning. Dotted lines indicate the performance of prior methods. (b) Performance comparison between SFT and RL policies under increasing training cost.

from embodied interactions in simulation. Recent studies on scaling laws (Kaplan et al., 2020) suggest that model performance improves predictably with increasing data volume, model size, and compute budget. However, the diminishing returns are exhibited as the system approaches the scaling frontier (Kaplan et al., 2020). As for the embodied navigation, this frontier arrives even earlier due to the relatively low-dimensional space of the action, the model quickly saturates its capacity to benefit from simply scaling in supervised learning. In large language models (LLMs), there have been numerous comparisons (Kumar et al., 2025) between two different paradigms for post-training: Reinforcement Learning (RL) and Supervised Fine-Tuning (SFT). However, it remains unclear how post-training should be approached in the field of robot learning, especially as more scalable training methods are being developed in this domain (Black et al.).

To systematically study this phenomenon, we first evaluate: 1) S2E-BC: A pure behavior cloning model trained solely on offline pretraining data. 2) S2E-SFT: An agent fine-tuned with data from URBAN-SIM (Wu et al., 2025) after pretraining. 3) S2E-Full: Our full approach combining pretraining followed by RL fine-tuning. Figure 7 (a) shows that increasing the data scale for S2E-BC yields only marginal improvements beyond a certain point. Expanding the dataset from 250k to 750k samples results in a mere 2% increase in success rate, suggesting that more offline data is insufficient to further enhance navigation capability. In contrast, RL significantly enhances the model's performance by leveraging interactions in simulation, achieving a 15% improvement in success rate over the pretrained model, with no additional offline data used. These results provide compelling evidence that RL can overcome the fundamental limitations of traditional scaling laws.

Additionally, we compare the performance scaling of supervised fine-tuning (SFT) and reinforcement learning (RL) across two benchmarks in Figure 7 (b), including the in-distribution test in URBAN-SIM and the out-of-distribution test in NavBench-GS. The data used for SFT is collected via a pretrained policy interacting within the environment, and episodes involving collisions are removed to ensure training stability. As additional training cost increases, RL maintains or improves success rates, while SFT suffers from severe overfitting. These results demonstrate that RL is not only more sample-efficient but also more robust under increased training costs.

| Method | Video Data | Empty | | | Obstacle | | | Pedestrian | | | Obstacle + Pedestrian | | |
|---|---|---|---|---|---|---|---|---|---|---|---|---|---|
| | | SR ↑ | RC ↑ | CT ↓ | SR ↑ | RC ↑ | CT ↓ | SR ↑ | RC ↑ | CT ↓ | SR ↑ | RC ↑ | CT ↓ |
| GNM | 70h | 0.23 | 0.51 | 0.72 | 0.16 | 0.49 | 0.90 | 0.09 | 0.53 | 1.28 | 0.07 | 0.44 | 2.31 |
| ViNT | 80h | 0.28 | 0.51 | 0.60 | 0.13 | 0.46 | 1.21 | 0.07 | 0.48 | 1.22 | 0.08 | 0.39 | 1.99 |
| NoMaD | 100h | 0.15 | 0.46 | 0.35 | 0.11 | 0.44 | 0.89 | 0.09 | 0.48 | **0.68** | 0.08 | 0.42 | 1.83 |
| MBRA | 700h | 0.61 | 0.75 | 0.35 | 0.51 | 0.77 | **0.53** | 0.71 | **0.84** | 1.01 | **0.51** | 0.69 | 2.09 |
| CityWalker | 2000h | 0.66 | 0.72 | 0.42 | 0.43 | 0.63 | 0.74 | 0.56 | 0.66 | 1.04 | 0.37 | 0.62 | 2.25 |
| CityWalker* | 100h | 0.67 | 0.90 | 0.15 | 0.52 | 0.79 | 1.00 | 0.63 | 0.66 | 2.34 | 0.47 | 0.51 | 2.52 |
| S2E | 100h | **0.82** | **0.92** | **0.00** | **0.57** | **0.78** | 0.69 | **0.74** | 0.78 | 1.50 | **0.51** | **0.73** | **1.58** |

Table 1: **NavBench-GS Benchmark.** Comparison of navigation foundation models across four tasks.

## 4.2 NAVBENCH-GS BENCHMARK

Although training durations and offline data sources vary between foundation models, our NavBench-GS benchmark remains fair and meaningful. We follow standard evaluation practices where foundation models are frequently trained on different corpora due to concerns about data privacy and distribution bias. We keep all environments unseen and distributionally distant from the training data.

**Benchmark.** The core and foundational function for a robot in urban scenarios is to navigate from point A to point B. In this task, the main commands beyond reaching the goal include: 1) not collide with static objects, and 2) not crash with moving objects. To this end, we design a benchmark in photo-realistic and physically interactive scenarios from Vid2Sim (Xie et al., 2025), spanning 26 scenarios, each instantiated with four tasks, *i.e.*, 1) empty environments, 2) environments with random static obstacles, 3) environments with moving pedestrians, and 4) environments with obstacles and pedestrians. We use success rate (SR), route completion (RC) and collision times (CT) to measure the model performance. An episode is deemed successful if the robot reaches the destination with a remaining distance of less than 1 meter and has fewer than 3 collisions with obstacles or pedestrians.

**Baselines.** To thoroughly evaluate our method's advantages, we compare against several state-of-the-art navigation foundation models: 1) image-based approaches, including GNM (Shah et al., 2023a), ViNT (Shah et al., 2023b), NoMaD (Sridhar et al., 2024), and 2) point-based approaches MBRA (Hirose et al., 2025), CityWalker (Liu et al., 2024b), CityWalker* (re-trained with the same dataset used in the current work). Several S2E variants are also evaluated in Table 4.

**Quantitative Results.** As demonstrated in Table 1, S2E consistently outperforms all baseline methods in both SR and RC across all test scenarios, validating the effectiveness of our S2E framework. Compared with point-based approaches, our results show that scaling performance with RL is significantly more effective than simply scaling up the amount of training data. For example, compared with CityWalker, our model trained with only 100h of data already surpasses video-based methods trained with over 2000h of data, underscoring the superior efficiency of reinforcement learning in scaling performance.

## 4.3 REAL-WORLD EVALUATIONS

For real-world evaluation, we use 25 real-world scenarios, each repeated 5 times to test the model's performance. We consider two types of environments: Empty, where only structural boundaries such as walls are present except for the ground; Obstacle, where static objects are randomly placed between the agent's starting point and the destination. We validate our approach through a comprehensive real-world evaluation using two distinct robotic platforms: 1) the Unitree GO2 quadrupedal robot, and 2) a wheeled robot. Figure 8 provides a qualitative comparison between S2E-Full and baseline methods, clearly demonstrating S2E-Full's superior collision avoidance capability in complex scenarios where other methods fail. Quantitative results in Table 2 further confirm these observations, with S2E-Full achieving the highest performance in both success rate and collision avoidance metrics. These results clearly illustrate that the interactive capabilities learned through RL training in simulation transfer effectively to the real world in a zero-shot manner. And as illustrated in Figure 9, the model demonstrates effective road keeping and collision avoidance when navigating where there are obstacles and pedestrians in the environment.

| Wheeled robot | | | |
|---|---|---|---|
| Method | SR ↑ | RC ↑ | CT ↓ |
| NoMaD | 0.25 | 0.55 | 0.76 |
| CityWalker | 0.28 | 0.44 | 0.78 |
| S2E-BC | 0.32 | 0.51 | 0.78 |
| **S2E-Full** | **0.51** | **0.64** | **0.60** |
| Quadruped robot | | | |
| Method | SR ↑ | RC ↑ | CT ↓ |
| NoMaD | 0.26 | 0.52 | 0.75 |
| CityWalker | 0.31 | 0.54 | 0.79 |
| S2E-BC | 0.34 | 0.63 | 0.91 |
| **S2E-Full** | **0.55** | **0.69** | **0.62** |

Table 2: **Real-world results.**

Figure 8: **Visualization of real-world results.**

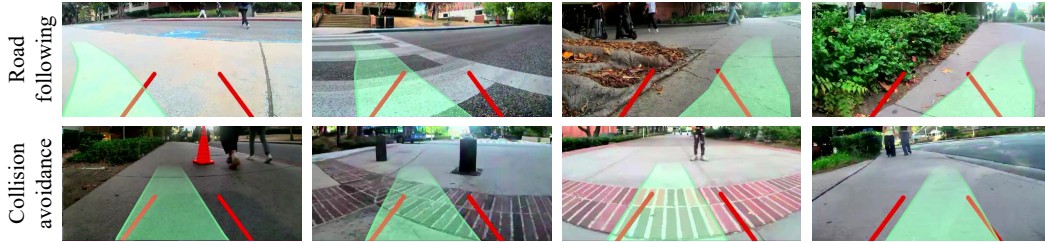

Visualization of S2E inference results on urban scenarios

Figure 9: **Deployment of S2E on real-world sidewalks.** We deploy S2E in real-world urban scenarios and visualize model predictions. Red lines denote the left and right sides of the wheeled robot at this moment, while the green bar indicates the predicted future trajectory. Note that the predicted future trajectories exhibit the capability of our model to follow sidewalks and avoid obstacles.

## 4.4 ABLATION STUDY

In this section, we conduct ablation studies to evaluate the effectiveness of key components of our method, particularly the AGDM and RAM for scaling with RL. More experiments are in the App. D.

**Effectiveness of anchor guidance.** As shown in App. D.5, S2E-BC-Single is trained with single-mode matching, while S2E-BC adopts anchor-guided distribution matching to model multi-modal distribution. Under the same setting, S2E-BC significantly outperforms S2E-BC-Single in both success rate (+11%) and collision rate (-0.64) in scenarios with obstacles, demonstrating that anchor-guided distribution matching improves the model's ability to capture complex distributions.

**Effectiveness of residual attention module.** To evaluate the proposed learning under the limited-module setting, we conduct ablation studies on different fine-tuning strategies, where DecFT-RL indicates fine-tuning on action decoder layers from pretrained initialization. As shown in Table 9, our approach achieves the highest success rate and lowest collision times on NavBench-GS-Obstacle, demonstrating the effectiveness of our finetuning strategy under limited-module adaptation. More results are provided in App. D.3.

| Methods | SR ↑ | CT ↓ |
|---|---|---|
| PPO | 0.02 | 2.37 |
| SFT | 0.49 | 0.77 |
| DecFT-RL | 0.39 | 0.91 |
| **Ours** | **0.57** | **0.69** |

Table 3: **Effectiveness of RAM.**

## 5 CONCLUSION

We propose a novel Seeing-to-Experiencing (S2E) framework for learning navigation foundation models. It integrates an Anchor-Guided Distribution Matching strategy to adapt to diverse real-world conditions and a Residual-Attention Module (RAM) for incremental improvement in interactive learning. Extensive experiments demonstrate that the models trained from our framework achieve zero-shot transfer to unseen scenarios and can be seamlessly deployed across different robots.

**Limitations and Future Work.** Since current systems lack 3D perception, even S2E sometimes fails to avoid collisions, which remains a persistent challenge for vision-only navigation approaches. One potential solution is integrating depth or occupancy prediction to infer 3D structural cues.

## ACKNOWLEDGMENT

The project was supported by the NSF Grants CNS-2235012, IIS-2339769, and TI-2346267. Honglin He is supported by the Amazon Trainium Fellowship. We thank Coco Robotics for the generous equipment donation.

## ETHICS STATEMENT

This work adheres to the ICLR Code of Ethics. All datasets used in this study are publicly available or derived from open-source simulation platforms; no private or sensitive data are involved. No sensitive data were collected during the experimental process. The research does not involve human or animal subjects. For real-world deployment, the robots operated under strict safety constraints: their maximum velocity and acceleration were limited by both hardware and software, and all tests were conducted in controlled environments. Researchers were present during experiments, with the ability to immediately intervene and stop the robot to ensure safety. The methods are designed to improve navigation safety and reliability, and are not intended for harmful applications. All experiments comply with institutional safety regulations and legal requirements. We are committed to research integrity, transparency, and reproducibility, and plan to release relevant code and model weights to facilitate verification by the community.

## REPRODUCIBILITY STATEMENT

We have made extensive efforts to ensure the reproducibility of our work. The main paper clearly specifies the model architecture ( App. E.2), training objectives (Section 3), and evaluation metrics (Section 4.2). Additional implementation details hyperparameter configurations, and environment settings are provided in the ( App. E). We plan to release them after double-blind review to enable full verification by the community. All datasets used in this study are publicly available, and preprocessing steps are described in ( App. E).

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

APPENDIX

## A  STATEMENT ON LARGE LANGUAGE MODEL USAGE

We used GPT-5&4o large language model (LLM) as a writing assistant to polish sentences, refine word choices, and check grammar consistency. The LLM was not involved in any reference collection, research ideation, experiment design, data analysis, or result interpretation. All technical contributions and conclusions in this work are solely the responsibility of the authors.

## B  DEMONSTRATION VIDEO

**We highly recommend watching our supplementary video for detailed demonstrations.** It presents a variety of experiments in both the simulator and the real world that thoroughly evaluate our S2E model across various settings and embodiments. *All real-world experiments conducted on different robots and scenes used the same S2E model.* The video consists of four sections:

1) S2E Capabilities Demonstration: highlights the ability of S2E in zero-shot deployment, including obstacle avoidance, interaction with pedestrians.

2) Long-horizon navigation: demonstrates the robustness of S2E in long-horizon urban environments.

3) Comparison with SOTA Methods: provides evaluations against representative baselines in real-world, demonstrating the performance of our method.

4) Data and Benchmark: introduces our training dataset for model pretraining, the interactive simulator used for policy finetuning, scenario overview of the NavBench-GS benchmark.

## C  REAL-WORLD EXPERIMENTAL RESULTS

In this section, we present experimental results in real-world scenarios, showcasing the zero-shot deployment performance of S2E. We first describe the setup and analysis of static obstacle evaluation in C.1, followed by experimental results involving dynamic human interactions with robots in C.2.

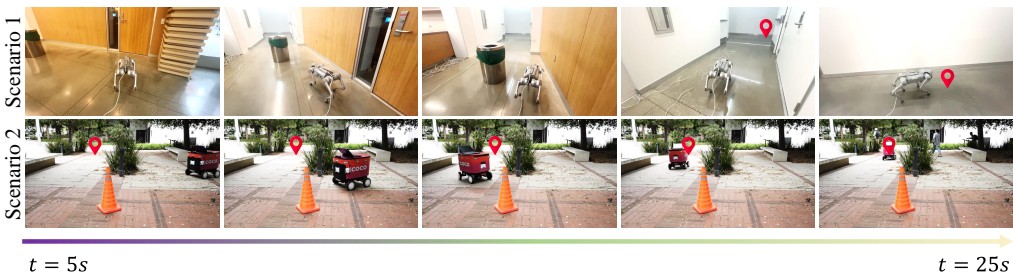

Figure 10: **Evaluation on scenarios with static obstacles**.

### C.1  SCENARIOS WITH STATIC OBSTACLES

We placed objects along the robot's path, such as a rubbish bin on the first row and a cone on the second row of Figure 10, to evaluate obstacle avoidance. Figure 10 demonstrates the performance of our S2E model in static obstacle scenarios: Scenario 1 illustrates GO2's navigation in an indoor environment, while Scenario 2 shows COCO's performance in an outdoor park setting with benches. The results confirm that our method reliably avoids obstacles.

### C.2  SCENARIOS WITH MOVING PEDESTRIANS

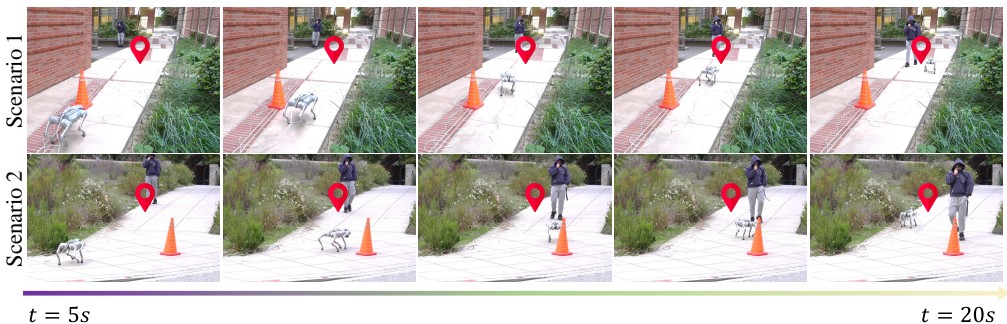

Figure 11: **Evaluation on scenarios with moving pedestrians**.

We designed these experiments to evaluate robotic navigation capabilities in dynamic environments. In addition to static obstacles (*e.g.*, the cone shown in Figure 11), we introduced a dynamic scenario where a pedestrian walks toward the GO2 robot, forcing it to replan its path. As demonstrated in Figure 11, when the pedestrian obstructs the robot's intended path, the robot dynamically adjusts its trajectory while still reaching the target destination.

## D  ADDITIONAL EXPERIMENTAL RESULTS

In this section, we present additional experiments to further analyze the framework design and effectiveness of S2E. We begin by evaluating different variants of S2E to highlight the effectiveness of our design D.1, followed by validating the effectiveness of reinforcement learning in improving

| Method | Empty | | | Obstacle | | | Pedestrian | | | Obstacle + Pedestrian | | |
|---|---|---|---|---|---|---|---|---|---|---|---|---|
| | SR ↑ | RC ↑ | CT ↓ | SR ↑ | RC ↑ | CT ↓ | SR ↑ | RC ↑ | CT ↓ | SR ↑ | RC ↑ | CT ↓ |
| S2E-Discrete | 0.44 | 0.80 | 0.07 | 0.37 | 0.74 | 0.92 | 0.40 | 0.74 | 1.36 | 0.35 | 0.53 | 2.01 |
| S2E-Diffusion | 0.65 | 0.86 | 0.03 | 0.44 | 0.75 | 0.91 | 0.60 | 0.68 | 1.43 | 0.37 | 0.66 | 2.07 |
| S2E-BC | 0.65 | 0.88 | 0.08 | 0.42 | 0.71 | 0.87 | 0.63 | 0.74 | 1.61 | 0.40 | 0.70 | 2.22 |
| S2E-PPO | 0.15 | 0.27 | 0.93 | 0.02 | 0.10 | 2.37 | 0.04 | 0.12 | 1.94 | 0.01 | 0.08 | 4.94 |
| **S2E-Full** | **0.82** | **0.92** | **0.00** | **0.57** | **0.78** | **0.69** | **0.74** | **0.78** | **1.50** | **0.51** | **0.73** | **1.58** |

Table 4: **NavBench-GS Benchmark.** .

other state-of-the-art methods (Shah et al., 2023b) in D.3. We then measure the generalizability of the model across diverse embodiments in D.4, conduct ablations on anchor points used for prediction in D.5. Furthermore, we provide qualitative results on the NavBench-GS benchmark in D.2.

## D.1 COMPARISON OF S2E VARIANTS

As shown in Table 4, we first compare S2E-DISCRETE (trained with classification on anchor points) and S2E-DIFFUSION (replace the Transformer decoder by DiT architecture as in DiffusionDrive (Liao et al., 2025)) against the baseline S2E-BC. The results demonstrate that the discrete formulation struggles to capture multimodality, leading to poor performance. Moreover, diffusion does not yield additional gains, as the GMM-based formulation is already sufficiently expressive to fit the data, while also being more amenable to reinforcement learning fine-tuning. The performance gains are significant when comparing S2E-Full to S2E-BC, with success rate improvements of 15% in obstacle settings, 11% in pedestrian scenarios, and 11% in obstacle-pedestrian scenarios. These substantial improvements across increasingly complex environments demonstrate RL's critical role in enhancing a policy's interactive capabilities.

## D.2 QUALITATIVE RESULTS ON NAVBENCH-GS BENCHMARK

To enable fair comparison with existing navigation models, we developed the NavBench-GS benchmark, which supports closed-loop evaluation for all navigation policies. Figure 12 presents representative test scenarios from our benchmark. Since these scenes typically lack roadside elements or interactive objects, we construct compositional scenes by adding static obstacles and dynamic pedestrians to increase the complexity and realism of the benchmark scenarios.

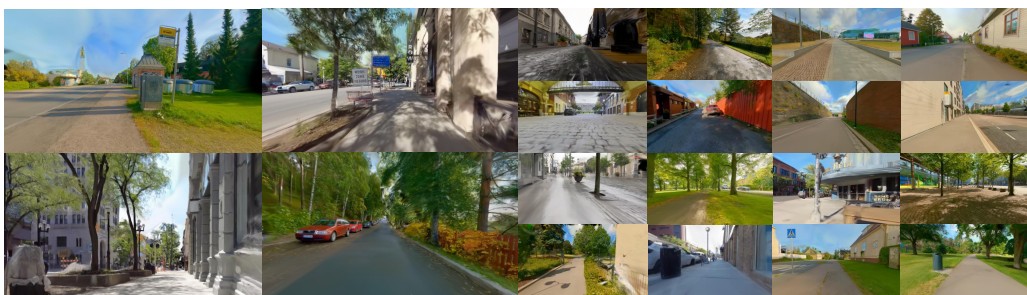

Figure 12: **Scenes in NavBench-GS Benchmark.**

Figure 13 shows compositionally rendered results on the NavBench-GS benchmark. We construct NavBench on top of URBAN-SIM (Wu et al., 2025), and employ instance-mask–based compositional rendering to seamlessly integrate backgrounds generated from 3D Gaussian Splatting (3DGS) Kerbl et al. (2023) with obstacles and pedestrians from the simulator. This design enables realistic scene synthesis and supports the evaluation of physical interactions in complex navigation environments.

Figure 14 shows comparisons among different variants of S2E in scenes NavBench-GS benchmark. In each scenario, we give the observation and the trajectory of the agent, along with the starting and the goal point. Colored trajectories correspond to different policies—S2E-PPO (red), S2E-BC under two different control schemes (pink and yellow), and the complete S2E-Full model (cyan). In both scenarios S2E-PPO and S2E-BC either deviate from the intended path or fail to reach the destination,

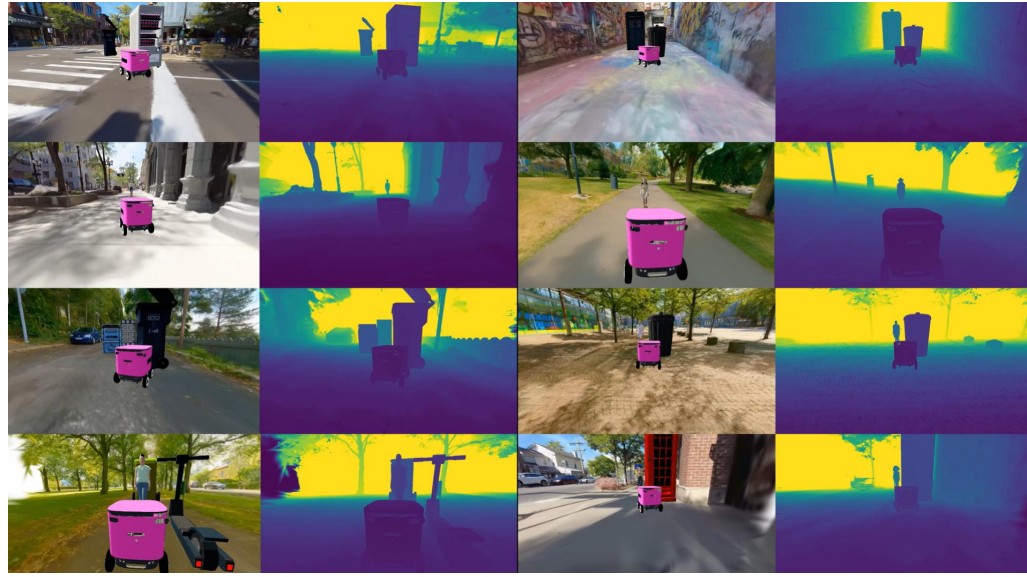

Figure 13: **Physical interaction in NavBench-GS Benchmark.**

becoming trapped by roadside obstacles or crashing into open space. By contrast, the S2E-Full successfully navigates around obstacles and converges to the goal in all cases, demonstrating its robustness.

As illustrated in Figure 15, column 1 shows our compositional scenes, where obstacles and pedestrians are overlaid onto base GS environments to simulate complex real-world navigation challenges. Column 2 and 3 demonstrate that NoMaD (Sridhar et al., 2024) becomes trapped by roadside obstacles (*e.g.*, traffic lights) or deviates from the intended path, while Citywalker (Liu et al., 2024b) successfully reaches the destination in Scenario 2. Notably, column 4 demonstrates that our method outperforms both approaches, achieving robust navigation in all scenarios containing both static and dynamic obstacles.

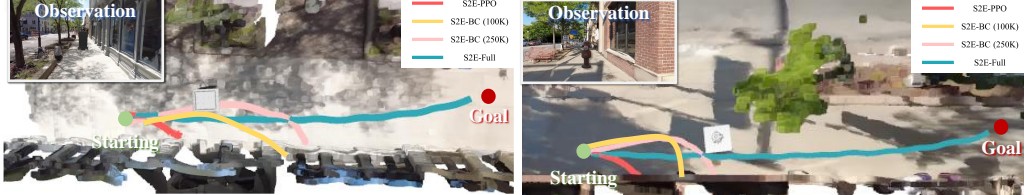

Figure 14: **Navigation Trajectories in NavBench-GS Benchmark.**

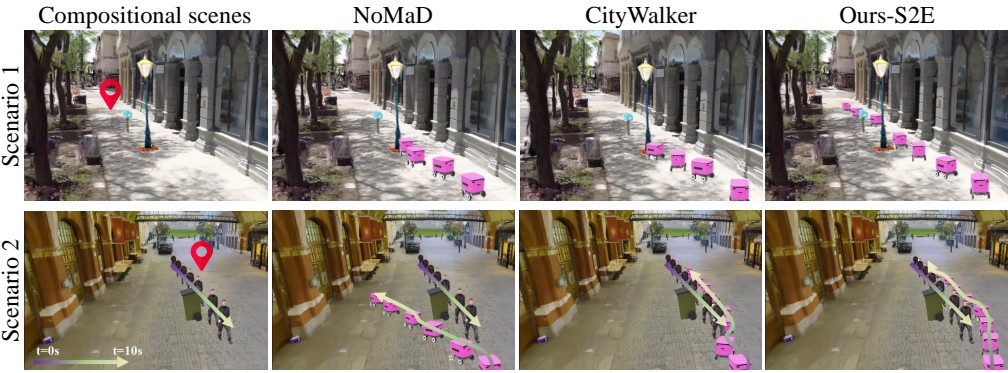

Figure 15: **Comparison with SOTAs on NavBench-GS Benchmark.**

### D.3 EFFECTIVENESS OF REINFORCEMENT LEARNING

To validate the effectiveness of RL in improving performance, we conduct more studies between the pretrained model and its finetuned weights in the simulator. Specifically, we further conducted experiments on RL-based finetuning on top of ViNT* (Shah et al., 2023b) introduced in manuscript, where only the policy head is updated while the backbone remains fixed. The setting of simulation environments, reward design, curriculum, etc. are consistent with the one used in S2E. As shown in results on Table 5, the finetuned policy outperforms the pretrained ViNT* across all evaluation metrics. Notably, it achieves significantly higher success rates and route completion, while also reducing collision cost. These results highlight the benefit of leveraging RL to continuously scale navigation foundation models that are trained solely on offline data.

| Method | SR ↑ | RC ↑ | CT ↓ |
|---|---|---|---|
| ViNT* | 0.27 | 0.31 | 2.01 |
| ViNT*+RL | **0.39** | **0.55** | **1.41** |

Table 5: **Performance comparison between pretrained and RL-finetuned on ViNT* (Shah et al., 2023b).** Finetuning the policy using RL significantly improves navigation performance across all metrics, which demonstrates the effectiveness of reinforcement learning in scaling navigation performance.

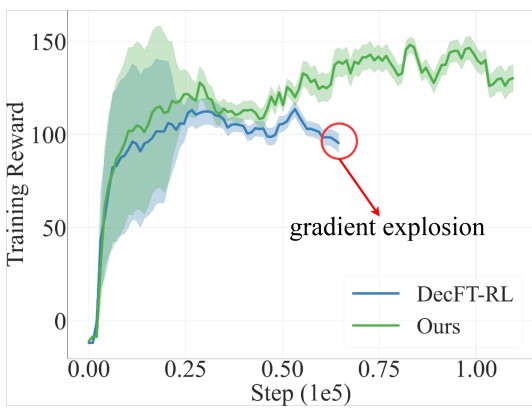

Figure 16: **Training curves of RL-finetuing on action decoder layers V.S. Ours- RAM.**

Additionally, we provide training curves comparing DecFT-RL (reinforcement learning fine-tuning applied directly to the action decoder layers) with our method. As shown in Figure 16, DecFT-RL quickly suffers from gradient explosion, whereas our approach enables stable training. Furthermore, during training, DecFT-RL requires nearly 40GB GPU memory, while our method only consumes 37GB on a single GPU with 64 parallel RL environments.

### D.4 CROSS-EMBODIMENT GENERALITY

| Robot Type | URBAN-SIM-Empty | | | NavBench-GS-Empty | | |
|---|---|---|---|---|---|---|
| | SR ↑ | RC ↑ | SPL ↑ | SR ↑ | RC ↑ | SPL ↑ |
| Wheeled Robot | **0.99 ± 0.06** | **0.99 ± 0.09** | 0.81 ± 0.26 | **0.92 ± 0.03** | **0.96 ± 0.02** | **0.90 ± 0.03** |
| Quadruped Robot | 0.93 ± 0.18 | 0.96 ± 0.10 | **0.91 ± 0.17** | 0.89 ± 0.08 | 0.89 ± 0.13 | 0.87 ± 0.08 |
| Humanoid Robot | 0.40 ± 0.16 | 0.75 ± 0.19 | 0.37 ± 0.19 | 0.21 ± 0.04 | 0.92 ± 0.09 | 0.18 ± 0.03 |

Table 6: **Cross-embodiment generality.** As a general navigation model, S2E can be directly deployed on various robotic platforms without any modifications.

Generalization across different embodiments is important for deploying or transferring policies in real-world applications. It would significantly reduce the retraining cost and improve the scalability of the model. To valid this capability, we evaluate the policy across three types of embodiments,

*i.e.*, wheeled, quadruped and humanoid robots. We test two scenarios: simulation scenes generated by URBAN-SIM and gaussian splatting scenes from NavBench-GS. As shown in Table 6, our approach maintains its performance across these different embodiments, showcasing the embodiment-agnostic property of the model. The wheeled and quadruped robots achieve high success rates, route completion, and SPL scores in all scenarios, demonstrating effective generalization across action controllers. While the humanoid robot shows lower success rate, its performance remains reasonable considering the increased joints. **Notably, all evaluations on both simulated and real-world environments across all robots use one S2E model**. We highly recommend referring to Section B and our demonstration video for more qualitative results and in real environments.

### D.5 ABLATION ON ANCHOR-BASED DISTRIBUTION MATCHING

Here, we provide the visualization results of the learned anchor points, as shown in Figure 17.

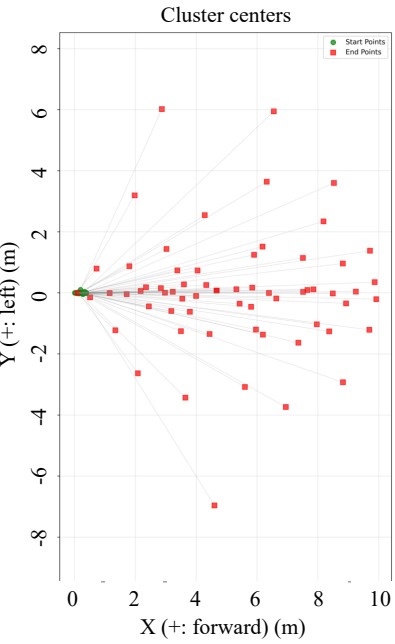

Figure 17: **Visualization results of anchor points from K-means.** Red squares indicate the clustered endpoints in the robot frame, while green circles denote the corresponding start-points in the robot frame.

We further investigate the influence of the number of anchors used in the S2E model. To systematically study its impact, we conduct an ablation study on the RECON (Shah et al., 2021) testset by varying the number of anchor points used during both training and inference. All experiments share the same visual encoder, training configuration, and evaluation protocol to ensure fair comparison.Anchor points serve as intermediate spatial targets that guide the policy toward the goal. As shown in Table 7, increasing the number of anchors from 1 to 64 improves performance, reducing minADE and improving mAP.

| # Anchor | minADE ↓ | mAP ↑ |
|----------|----------|-------|
| 1        | 0.21     | 0.57  |
| 4        | 0.17     | 0.58  |
| 8        | 0.13     | 0.62  |
| 16       | 0.13     | 0.59  |
| 64       | **0.09** | **0.69** |

Table 7: **Ablation on anchor point number.** We vary the number of anchor points used in the S2E model and evaluate on the RECON (Shah et al., 2021) testset. Increasing the number of anchors improves performance up to a point, with 64 anchors yielding the best results.

| # Anchor | SR ↑ | CT ↓ |
|---|---|---|
| S2E-BC-Single | 0.33 | 1.51 |
| S2E-BC | 0.42 | 0.87 |

Table 8: **Comparison on NavBen-GS-Obstacle.**

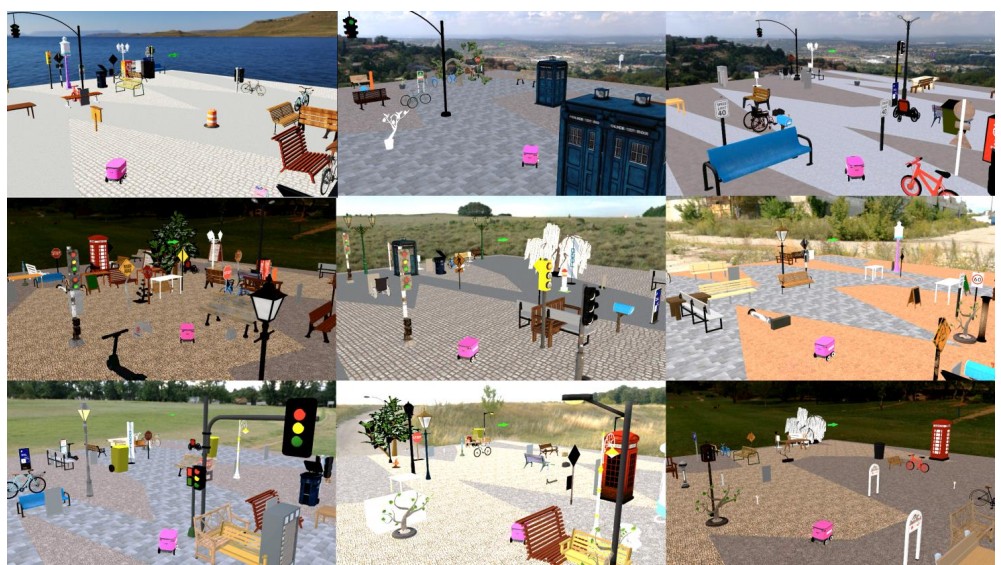

Figure 18: **Examples of simulation environments in URBAN-SIM for RL finetuning.**

As shown in Table 8, we compare variants of S2E-BC with different anchors on the NavBench-GS-Obstacle benchmark. When using only a single mode for regression (S2E-BC-SINGLE), the model often fails to capture multimodal behaviors and thus exhibits significantly lower success rate and higher collision time. In contrast, employing multiple anchor modes (S2E-BC) provides a richer representation space, leading to improved success rate and reduced collision time.

### D.6 ABLATION ON SIMULATION ENVIRONMENTS

In the default setting of URBAN-SIM (Wu et al., 2025), RL environments are randomly sampled across the region, which deviates from realistic urban layouts. To address this, we redesign the environment layout with a new set of procedural generation rules (to be released as open source), enabling the creation of more structured and realistic simulation scenarios, as illustrated in Figure 18 and Figure 19. As shown in Table 9, our redesigned environment distribution leads to notable improvements in both success rate and collision time, demonstrating the effectiveness of spatially coherent RL environments.

| Methods | SR ↑ | CT ↓ |
|---|---|---|
| S2E-UrbanSim | 0.47 | 0.74 |
| **Ours** | **0.57** | **0.69** |

Table 9: **Effectiveness of the spatial distribution of RL envs.**

## E EXPERIMENTAL DETAILS

In this section, we provide more details regarding the model architecture, dataset and simulation environments, training strategies, and robots used in our experiments. We begin with a description of the datasets and simulation environments in E.1, followed by the S2E model structure in E.2. E.3 and E.4, E.5 describe the pretraining and finetuning strategies, respectively. Finally, Sections E.6 and E.7 introduce the simulated and real-world robotic platforms.

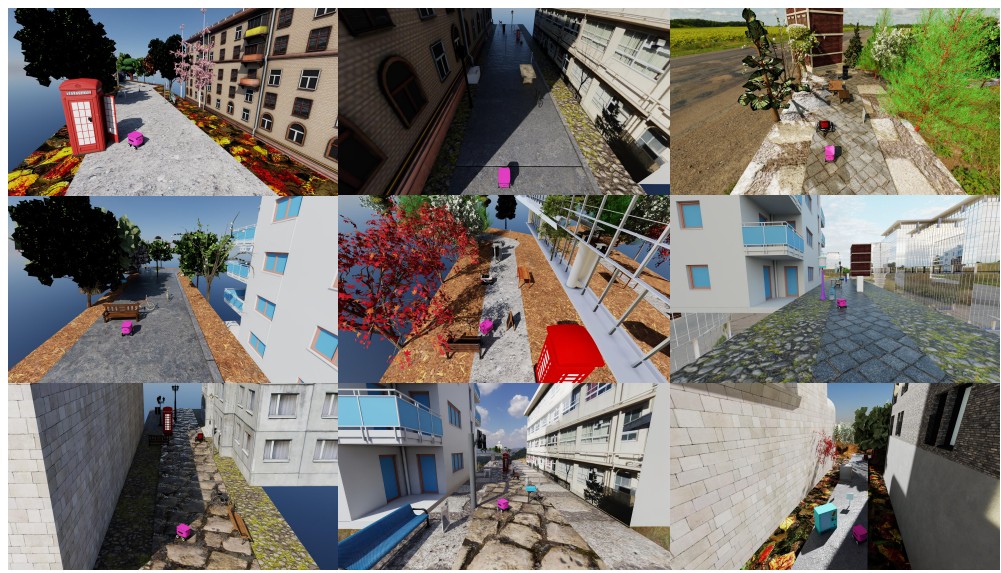

Figure 19: **Examples of simulation environments in our experiments for RL finetuning.**

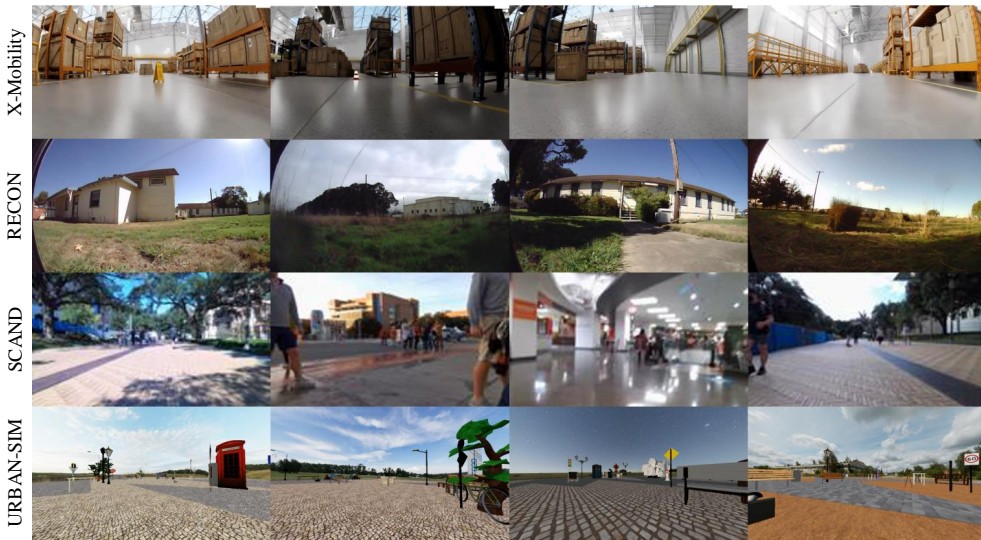

Figure 20: **Examples of dataset for pretraining.**

E.1    DETAILS OF DATASET AND ENVIRONMENTS

The S2E training dataset contains over 100 hours of navigation trajectories, sourced entirely from existing real-world records (Shah et al., 2021; Karnan et al., 2022b) or simulation platforms (Liu et al., 2024a; 2025; Wu et al., 2025). The dataset consists of a combination of navigation behaviors collected across 6 distinct robotic platforms, as shown in Table 10 and Figure 20. For real-world datasets, trajectories are labeled using odometry or fused GPS/IMU estimations. In simulation environments, ground-truth poses are directly extracted from the simulator. All data are converted into standardized observation-action pairs by converting applying egocentric transformation.

As shown in Figure 19, we use a diverse set of procedurally generated simulation environments from URBAN-SIM (Wu et al., 2025) and modified the procedural generation pipeline for finetuning. These environments are designed for urban navigation, including static obstacles and varying scene layouts. The diversity and randomness in object placement, texture, and lighting encourage robust policy finetuning during reinforcement learning.

| # | Dataset | Platform | Hrs. Used | Environment |
|---|---------|----------|-----------|-------------|
| 1 | RECON (Shah et al., 2021) | Jackal | 25h | off-road |
| 2 | SCAND (Karnan et al., 2022b) | Spot, Jackal | 5h | sidewalks |
| 3 | FrodoBots-2K (Team, 2024; Hirose et al., 2025) | FrodoBot | 65h | sidewalks |
| 4 | X-Mobility (Liu et al., 2024a; 2025) | Nova Carter | 2.5h | warehouse |
| 5 | URBAN-SIM (Wu et al., 2025) | COCO, GO2, G1 | 2.5h | sidewalks |

Table 10: **Overview of Training Datasets.** The table summarizes the robot platforms, environments and useful details covered in each dataset used in S2E pretraining stage.

### E.2 DETAILS OF MODEL ARCHITECTURE

In this section, we provide details of the model architecture used in S2E. We adopt a DINOv3-based visual encoder (Siméoni et al., 2025). Specifically, we first encode the past 10 frames sampled at 5Hz into frame-level tokens, and encode the current frame into path-level tokens via a grid pooling strategy, resulting in a total of $10 + 64$ tokens. The goal point is embedded through a lightweight linear MLP, while the goal image is encoded using DINOv3, and the token length of goal image is 16 after grid pool. These tokens are then combined with anchor representations and processed by a Transformer (Vaswani et al., 2017)-based architecture consisting of a 6-layer encoder and a 6-layer decoder, each with 8 attention heads and a feedforward dimensionality of 3072 (*i.e.*, $4 \times 768$). The encoder operates over the observation and goal tokens, while the decoder takes observations and goal as keys $K$ and values $V$, and anchors as queries $Q$, to generate action-relevant representations. Finally, the anchor features are decoded by three separate MLP heads to predict normalized trajectories with velocity and score. To mitigate shortcut learning, we apply stochastic masking over the goal signals: with a probability of 0.35 neither the goal image nor the goal point is provided, with a probability of 0.20 only the goal image is available, with a probability of 0.40 only the goal point is available, and with a probability of 0.05 both signals are provided.

### E.3 DETAILS OF PRETRAINING

As shown in Table 11, we provide a detailed list of hyperparameter used in pretraining stage of S2E.

| S2E Model | | S2E Pretraining | |
|-----------|------|-----------------|------|
| RGB Resolution | $256 \times 256$ | # Epochs $n_{ep}$ | 100 |
| Encoder | Dinov3 | Batch Size | 256 |
| Token Dimension | 768 | Learning Rate | $2 \times 10^{-4}$ |
| Attention Hidden Dimension | 3072 | Optimizer | AdamW |
| # Attention Layers $n_L$ | 6 | LR Schedule | Cosine |
| # Attention Heads $n_H$ | 8 | Scheduler Period | 40 |
| Temporal Context $k$ | 10 | Compute Resources | $8 \times$ NVIDIA L40S |
| Prediction Horizon $T$ | 10 | | |

Table 11: **Architectural and Pretraining Hyperparameters for S2E.** Left: model architecture. Right: training configuration.

### E.4 DETAILS OF FINETUNING

In this section, we first introduce the finetuning strategy for the GMM-based policys.

**Standard deviation reinitialization.** We observed empirically that the action distribution learned from imitation learning tends to have a very small standard deviation (*i.e.*, $\sigma \to 0^+$), which hampers sufficient exploration in reinforcement learning. However, retraining the action decoder is undesirable, as the anchor–action mapping has already been well captured by imitation learning. To address this, we introduce an additional head to generate the log standard deviation, initialized to zero (corresponding to $\sigma = 1$), thereby ensuring adequate exploration during RL fine-tuning.

We use the standard sampling strategy for GMM in RL finetuning. Given the mixture weights $\{q_i\}_{m=1}^M$, means $\{\mu_i\}_{m=1}^M$, and standard deviations $\{\sigma_i\}_{m=1}^M$, an action $a$ is sampled by first selecting

a component index $m \sim \text{Categorical}(q)$, then sampling from the corresponding Gaussian:

$$a \sim \mathcal{N}(\mu_m, \sigma_m^2), \tag{12}$$

Since the exact entropy of a GMM does not have a closed-form solution, we use a simplified estimation of its lower bound. First, we approximate it based on the statement of (Huber et al., 2008; Wang et al., 2024):

$$\mathcal{H}_\pi \approx \sum_{m=1}^M q_m \cdot \left[\tfrac{1}{2}\log\big((2\pi e)^2|\Sigma_m|\big)\right] - \sum_{m=1}^M q_m \log q_m, \tag{13}$$

$$\Sigma_m = \begin{bmatrix} \sigma_x^{m2} & \rho^m \sigma_x^m \sigma_y^m \\ \rho^m \sigma_x^m \sigma_y^m & \sigma_y^{m2} \end{bmatrix}, \tag{14}$$

we further set $\rho^m = 0$ during finetuning, so we have the estimation:

$$\mathcal{H}_\pi \approx \sum_{m=1}^M q_m \cdot \left[\tfrac{1}{2}\log\big((2\pi e)^2 \sigma_x^{m2} \sigma_y^{m2}\big)\right] - \sum_{m=1}^M q_m \log q_m. \tag{15}$$

We train our agent with 64 parallel environments, and the detailed list of hyperparameter used in finetuning stage of S2E is shown in Table 12.

| Hyperparameter | Value |
|---|---|
| # Epochs $n_{ep}$ | 300 |
| Minibatch Size | 512 |
| Learning Rate | $1 \times 10^{-5}$ |
| Optimizer | AdamW |
| LR Schedule | Adaptive |
| KL Threshold | 0.01 |
| Clip Range | 0.2 |
| GAE $\lambda$ | 0.95 |
| Discount Factor $\gamma$ | 0.99 |
| Entropy Coefficient | 0.001 |
| Rollout Horizon | 32 |
| Gradient Clipping | 1.5 |
| Value Loss Coefficient | 2.0 |
| Compute Resources | $1\times$ NVIDIA L40S |
| Finetuning Time | 8 hrs |

Table 12: **S2E PPO Finetuning Hyperparameters.** PPO-related training configurations used for finetuning the policy in our experiments.

### E.5 DETAILS OF REWARD FUNCTION DESIGNS.

We design the reward function $R$ as:

$$R = R_{\mathcal{G}} + R_{\mathcal{R}} + R_{\mathcal{H}}, \tag{16}$$

$$R_{\mathcal{G}} = R_{g,d} + R_{g,s} + R_{c,d} + R_{c,s}, \tag{17}$$

$$R_{\mathcal{R}} = R_{walkable}, \tag{18}$$

$$R_{\mathcal{H}} = R_{trajectory} + R_{heading}. \tag{19}$$

- **Dense goal-reaching reward** $R_{g,d}$: Defined as $R_{g,d} = 1 - \tanh(d_g/\sigma)$, where $\sigma$ controls the distance scale and $d_g$ denotes the distance to the goal point. We use two standard

deviations ($\sigma_{coarse} = 10, \sigma_{fine} = 2$) for coarse and fine levels of shaping. Additionally, a small velocity-alignment term is included, defined as $R_{vel} = 1 - \frac{\cos(\mathbf{v}, \mathbf{p}_{rel})}{\|\mathbf{p}_{rel}\|}$, which encourages the agent's velocity $\mathbf{v}$ to align with the relative pose $\mathbf{p}_{rel}$ toward the goal.

- **Sparse goal-reaching reward** $R_{g,s}$: A large terminal reward of $+2000$ is given upon successfully reaching the goal, and the episode would be terminated at the same time.

- **Dense collision-avoidance reward** $R_{c,d}$: Penalizes proximity to the nearest obstacle center $R_{c,d} = -0.001 \times \text{clip}(\frac{1}{d_o + 1e^{-5}}, 20)$, where $d_o$ denotes the distance to the nearest obstacle center.

- **Sparse collision-avoidance reward** $R_{c,s}$: A penalty of $-200$ is applied when a collision occurs, and the episode would be terminated at the same time.

- **Road-keeping reward** $R_{walkable}$: If the agent leaves the walkable area, the episode terminates with a penalty of $-10$.

- **Trajectory-similarity reward** $R_{trajectory}$: Penalizes deviation from the reference global trajectory, scaled as $-0.00005 \times \Delta_{\text{traj}}$.

- **Smoothness reward** $R_{heading}$: Penalizes abrupt heading changes, computed as the sum of incremental heading differences $\sum_t |\Delta\theta_t|$, scaled as $R_{heading} = -0.005 \times \sum_t |\Delta\theta_t|$.

## E.6 ROBOTS IN SIMULATOR

All simulated robotic platforms are built upon NVIDIA IsaacSim (Xu et al., 2022; Dorbala et al., 2023), a high-fidelity and GPU-accelerated simulation environment that supports physics-based interactions and photorealistic rendering. Each robot is modeled using official URDF descriptions and equipped with RGB cameras.

For locomotion, we employ a modular control stack that consists of a low-level joint controller and a high-level policy trained with reinforcement learning. The policy is trained using Proximal Policy Optimization (PPO) on terrain-randomized environments with curriculum learning. The reward function encourages stability, velocity tracking, and energy efficiency, while penalizing collisions and falls. Domain randomization in texture, friction, and mass distribution is applied to improve sim-to-real transferability. We adopt a shared framework across robot types, enabling scalable embodiment-aware training in simulation.

**Wheeled robot.** A differential-drive robot, controlled via a kinematic model (Polack et al., 2017), where linear and angular velocities $(v, \omega)$ (calculated from waypoint via a ideal PD controller (Sridhar et al., 2024)) are used as control commands. The simulator integrates these commands through rigid-body physics engine, with frictional contacts determining actual wheel-ground interaction.

**Unitree-GO2 quadruped robot.** A quadruped platform capable of versatile locomotion in complex terrains, the action is a 3-dimensional vector $(v_x, v_y, \omega)$ (calculated from waypoint via a ideal PD controller (Sridhar et al., 2024)). The locomotion model is a lightweight MLP trained based on the standard training environments provided by IsaacSim (Xu et al., 2022; Dorbala et al., 2023).

**Unitree-G1 humanoid robot.** A humanoid agent with articulated head, torso, and leg joints excluding hand actuation, the action is a 3-dimensional vector $(v_x, v_y, \omega)$ (calculated from waypoint via a ideal PD controller (Sridhar et al., 2024)). The locomotion model is a lightweight MLP trained based on the standard training environments provided by IsaacSim (Xu et al., 2022; Dorbala et al., 2023).

## E.7 ROBOTS IN REAL WORLD

**Wheeled robot.** A differential-drive platform equipped with RGB cameras and differential drive odometry, deployed for sidewalk navigation. The robot is controlled via the same kinematic model as in simulation, where linear and angular velocities $(v, \omega)$ are computed from waypoints using an ideal PD controller. The odometry is used for real-time position estimation and providing target position during navigation continuously.

**Unitree-GO2 quadruped robot.** The real-world GO2 robot is equipped with RGB sensors and a low-level locomotion controller provided by Unitree. Instead of executing joint-level commands from a trained policy, we interface with the GO2 through its built-in velocity control API, sending

high-level commands $(v_x, v_y, \omega)$ to leverage its native gait generation and stability modules. There is a lidar-based odometry used for real-time position estimation and providing target position during navigation continuously.

