# OpenReview forum: "From Seeing to Experiencing: Scaling Navigation Foundation Models with Reinforcement Learning"
_ICLR.cc/2026/Conference — ICLR 2026 Poster_

### Official Review · Reviewer_UWpN · 2025-10-23

**Soundness:** 3
**Presentation:** 3
**Contribution:** 3
**Rating:** 6
**Confidence:** 5

**Summary:**

The paper proposes Seeing-to-Experiencing (S2E), a training recipe for goal-conditioned navigation that keeps visual priors from offline videos and adds interaction skills with reinforcement learning in simulation. It introduces a new action representation, using an anchor-conditioned Gaussian mixture to represent multiple short-horizon motions under the same observation. It also adds a Residual-Attention Module that fine-tunes only residual cross-attention while freezing visual encoders and self-attention to limit sim-to-real drift.
Experiments in the NavBench-GS benchmark show RL fine-tuning outperforms supervised fine-tuning under the same budget, scales better than more offline data alone, and transfers zero-shot to wheeled and legged robots, with ablations confirming both modules are important.

**Strengths:**

1. The paper shows reinforcement learning is extremely helpful for closed-loop navigation tasks.
2. The paper proposes using several fixed anchor points and GMM to represent the action space. This is essential specifically to the navigation task. The authors also provide in-depth analysis and comparison to other representations.
3. The paper provides insightful discovery on the data saturation margin in the embodied navigation task and shows RL to be a more robust and efficient learning technique.
4. The paper shows zero-shot depolyment of the trained model on both wheeled and quadroped robots. Experiment result shows superior performance compared to other baselines.

**Weaknesses:**

1. The "Residual attention module" paragraph in section 3.2 (Line 258-268) is a bit confusing. The statement is not a well established conclusion in literature. The authors should rewrite the paragraph by providing more concrete derivation of the conclusion.
2. In ablation study (Line 462-466 and Appendix Sec. D.5), there is no comparison with difussion policy models. The authors should add comparison to that to better support the discussion in Sec. 3.1.

**Questions:**

1. In Fig. 6(b) left, why does SFT still sufer from overfitting even in the in-distribution Urban-sim evaluation?
2. In reinforcement learning and simulative benchmark, there is no specific embodiment. Then how is collision checking done? Are they achieved by a navmesh or using a uniform collision volumn?
3. In Tab. 1, what does ZeroPolicy mean? Is it a pure RL model? I'm also curious on the authors insights on why (or why not) pure RL training might not perform well in urban navigation while they work well in indoor environments like the HM3D dataset?

---

> ### Author Response · Authors · 2025-11-24
> **Response to Reviewer UWpN - Part 1**
>
> We thank the reviewer for the constructive and insightful feedback. We are encouraged that the reviewer recognized the effectiveness of RL for closed-loop navigation ("extremely helpful"), the contribution of our anchor-guided GMM action representation, detailed comparisons, and "in-depth" analysis, as well as the model’s zero-shot performance. We address the concerns as follows.
>
> ## Weaknesses
> > ## Q1: Description of residual attention module.
>
> We thank the reviewer for this constructive suggestion. To address this, we have thoroughly revised **Section 3.2 (Page 5, 6)** to provide a more concrete derivation, theoretical and empirical insight of our design choices. The modifications are highlighted on **Pages 5 and 6**. Briefly, we: (1) added fundamental problems of full-parameter fine-tuning, (2) described the mechanics and optimization process of our approach to explain why it works.
>
> Here, we additionally provide the expanded results of Table 3 (on Navbench-GS-Obstacle), Reinforcement learning with full-parameter fine-tuning (FullFT-RL). Both DecFT-RL and FullFT-RL show clear performance degradation due to the instability of closed-loop fine-tuning, illustrating the difficulty of optimizing all parameters of a visual navigation model with RL. In contrast, our RAM achieves the highest success rate and the lowest collision count, demonstrating the advantage of parameter-efficient, interaction-focused adaptation.
>
> | Methods | SR $\uparrow$  | CT $\downarrow$ |
> | :--- | :---: | :---: |
> | PPO | 0.02 | 2.37 |
> | SFT | 0.49 | 0.77 |
> | DecFT-RL | 0.39 | 0.91 |
> | FullFT-RL | 0.31 | 1.33 |
> | **Ours** | **0.57** | **0.69** |
>
> > ## Q2: Comparison with difussion policy models.
>
> Thanks for the constructive and insightful comment. We have added comparisons with different variants of the S2E models, including versions where a diffusion policy is used as the action decoder. These results are provided on **Table 4 (Page 19)**. In addition, we include the performance of a pure [goal point based (goal point is optional)] diffusion-policy model implemented using the NoMaD codebase; the results are shown below.
>
> | Methods | SR $\uparrow$ | CT $\downarrow$ |
> | :--- | :---: | :---: |
> | S2E-BC | 0.42 | 0.87 |
> | S2E-Diffusion | 0.44 | 0.91 |
> | DiffusionPolicy | 0.32 | 0.93 |
> | **Ours** | **0.57** | **0.69** |
>
> Overall, although using a diffusion-based action head leads to minor improvements, diffusion policies are still not sufficiently stable for reinforcement learning in closed-loop settings. Because addressing the limitations of diffusion models within RL is not the central focus of this work, and maintaining training stability is essential for S2E, we adopt an MLP-based action head in our main model. Thank you again for the helpful suggestion and we will try to extend the RL framework with diffusion policy in future work.
>
> ## Questions
> > ## Q3: Overfitting issues of SFT.
>
> Thank you for this constructive comment. Even though the Urban-Sim evaluation is an in-distribution setting, SFT can still exhibit overfitting for two key reasons.
>
> First, SFT is trained purely on demonstrations collected under a set of trajectories (Detail process would be introduced later). For fair comparison across methods, we keep the scenario identical for SFT and RL, which means the model only sees a limited subset of possible navigation behaviors in limited scenarios.
>
> Second, despite being in-distribution, distributional shift still occurs during rollouts. SFT minimizes a per-step supervised loss, but during deployment the model actions can deviate slightly from the expert, leading it into states that are not well covered by the training data. This compounding error, a limitation of behavior cloning, results in decreased success rates even under in-distribution evaluation.

---

> > ### Author Response · Authors · 2025-11-24
> > **Response to Reviewer UWpN - Part 2**
> >
> > ### **Details of data collection process in Urban-Sim for SFT**
> >
> > | Stage                        | Description |
> > | ---------------------------- | ---------------------------------------------------------------------------------------------------------------------------------------------------------------------------------------------------------------- |
> > | **1. Rollout generation**    | We use the pretrained model to navigate in all Urban-Sim training scenarios. At each step, the model outputs an action distribution, and the executed action is sampled from this distribution.|
> > | **2. Closed-loop execution** | The robot follows the sampled actions to generate full navigation trajectories in a closed-loop manner. This produces realistic on-policy behaviors that reflect the pretrained decision-making pattern of model.|
> > | **3. Success filtering**     | Only trajectories that successfully reach the goal without collisions are kept. Failed rollouts (e.g., stuck, drifting, or colliding cases) are discarded to ensure clean and reliable supervision. |
> > | **4. Dataset construction**  | The remaining successful trajectories are aggregated into the SFT dataset, consisting of observation–action pairs of the form ((o_t, a_t)). These serve as expert demonstrations for supervised fine-tuning.|
> > | **5. SFT training**          | SFT minimizes a per-step supervised loss over this dataset, producing an imitation-based policy initialized from on-policy rollouts of the pretrained model.|
> >
> > > ## Q4: Details of collision checking in the benchmark.
> > Thank you for the insightful question. In our reinforcement learning and simulation benchmark, we use a **simplified wheeled robot model** in Urban-Sim. The agent is simulated with basic dynamics: its mass is randomly sampled from a predefined range, and a lightweight PD controller maps the predicted linear and angular velocities into low-level joint commands. These commands are applied using the simulator’s default controller. The exact controller parameters and sampling ranges will be provided in the released code. It is notable that the **same** trained model is deployed directly on both wheeled and quadruped robots without more embodiment-specific tuning, demonstrating that our policy generalizes across different platforms.
> >
> > > ## Q5: Details of ZeroPolicy and pure RL models.
> >
> > Thank you for the insightful question. Regarding ZeroPolicy, it is not a learned RL agent. ZeroPolicy is a simple handcrafted controller: given the target goal position, we compute the relative displacement in the robot frame and convert it into linear and angular velocity commands through a control rule. This baseline serves only as a sanity check for the evaluation metrics to reflect the inherent complexity of the scene, and it does not involve any learning. We also provide results for a pure RL model in **Table 4 (Page 19)**, trained from scratch on Urban-Sim. Pure RL fails to achieve meaningful performance. There are several reasons for this gap. Urban environments exhibit highly complex layouts, long-horizon navigation structure, and diverse obstacle geometry that the simulated scenarios cannot fully capture with sufficient coverage. As discussed in the revised **Section 3.2**, the visual domain in urban navigation is also significantly more variable than indoor datasets without strong pretrained visual priors. These factors combined make pure RL unreliable and prone to divergence in outdoor urban settings.

---

> > > ### Comment · Reviewer_UWpN · 2025-11-25
> > >
> > > I appreciate the authors and their effort in the rebuttal. All my concerns and questions are well addressed. I keep my original rating.

---

> ### Author Response · Authors · 2025-11-26
>
> Dear Reviewer UWpN,
>
> Thank you very much for your thoughtful feedback. We appreciate your time and effort in reviewing our work, and we will further revise the paper accordingly based on your valuable suggestions.
>
> Best regards,
>
> Authors of Paper #9621

---

### Official Review · Reviewer_FY2s · 2025-10-31

**Soundness:** 3
**Presentation:** 2
**Contribution:** 3
**Rating:** 4
**Confidence:** 4

**Summary:**

This paper proposes the S2E framework, combining offline video pretraining (via AGDM strategy) and simulation RL finetuning (with RAM module). It builds NavBench-GS, verifies RL boosts navigation model performance, breaks diminishing returns of data scaling, and enables zero-shot transfer across robots.

**Strengths:**

1. This paper proposes Anchor-Guided Distribution Matching (AGDM), which uses an anchor-guided Gaussian Mixture Model to model multimodal navigation trajectories, capturing diverse valid actions under the same observation while ensuring training stability.

2. This paper designs the Residual-Attention Module (RAM), which freezes pretrained components and adds trainable residual branches to cross-attention layers, enabling the model to gain interactive skills via RL without losing pretrained generalizable knowledge.

3. This paper establishes the NavBench-GS benchmark, built on photorealistic 3D Gaussian Splatting scenes with physical interactions, realizing closed-loop policy evaluation and solving the reproducibility issue of real-world navigation testing.

**Weaknesses:**

1. The model relies solely on visual input and lacks 3D perception capabilities, leading to occasional failure in obstacle avoidance in some scenarios, which is a persistent limitation for vision-only navigation approaches.

2. The real-world evaluation scenarios are relatively limited (only 25 scenarios), and the generalization performance of the S2E framework in more complex and diverse urban environments (e.g., extreme weather, complex traffic conditions) has not been verified.

3. The humanoid robot in cross-embodiment evaluations shows notably lower success rates compared to wheeled and quadruped robots, yet the paper does not deeply analyze the root causes (e.g., joint complexity impacts) or propose targeted optimization strategies for humanoid platform adaptation.

4. The RL finetuning relies on modified URBAN-SIM environments, but the paper only briefly mentions procedural generation rule adjustments without detailing how these rules ensure the simulated environments fully align with real urban spatial layouts, potentially limiting the sim-to-real transfer reliability

**Questions:**

1. Given that the current model lacks 3D perception, what specific 3D information integration methods (such as depth prediction or occupancy prediction) do you plan to adopt in future work, and how will you balance the computational cost of 3D perception with the real-time performance of navigation?

2. The NavBench-GS benchmark currently covers 26 scenarios, but real urban environments involve more dynamic elements (e.g., sudden appearance of vehicles, temporary road closures). Will you expand the benchmark to include such scenarios, and what criteria will be used to select new scenarios to ensure the benchmark’s representativeness?

---

> ### Author Response · Authors · 2025-11-24
> **Response to Reviewer FY2s - Part 1**
>
> We thank the reviewer for the constructive and insightful feedback. We are encouraged that the reviewer found our two main technical innovations, and “solving the reproducibility issue” of real-world navigation testing by estabilshing NavBench-GS benchmark. We address the concerns as follows. (We provide additional visualization results **[Anonymous Video Link](https://h1h2h3h4h5h.github.io/)** to address the concerns from the reviewer.)
>
> ## Weaknesses
> > ## Q1: The model relies solely on visual input and lacks 3D perception capabilities, leading to occasional failure in obstacle avoidance.
>
> Thank you for the insightful comment. In many practical applications, such as COCO Robotics, to reduce the cost, improve the speed of data stream, reduce the space for data storage, robots are equipped with RGB cameras only, without 3D sensing such as LiDAR or depth. And we focus on this real-world-existing RGB-only navigation setting.
> While integrating 3D perception would indeed improve navigation robustness, it is **beyond the scope of our current problem formulation**. But we totally agree that how to integrate 3D info could be an interesting future work.
>
>
> > ## Q2: Limited real-world evaluation results.
>
> While Table 2 reports 25 scenarios, each scenario includes **two variants** (clean and obstacle), and multiple trials for each variant. To further address the reviewer’s concern regarding the **generalization** and **robustness** of S2E, we provide additional videos demonstrating the model’s performance in more complex environments and extreme weather conditions at the following **[Anonymous Video Link](https://h1h2h3h4h5h.github.io/)**. These scenarios involve rainy-night conditions with low illumination, strong reflections, and headlight glare, or environments with complex obstacles and dense pedestrian interactions, demonstrating the effectiveness and the generalizability of our approach.
>
> > ## Q3: Humanoid platform adaptation.
>
> 1. **Controller limitation.**   For the humanoid, we use a lightweight MLP locomotion controller to map output from S2E to joint states. Failure cases mainly arise from this controller. To further improve the stability and smoothness of humanoid motion, we additionally implement an MPC controller that refines trajectory, reducing sudden changes and significantly improving tracking performance, and the controller would be made publicly available when releasing the code. We will add the latest results as soon as possible when they are ready.
>
>
> 2. **Lack of humanoid training data.** The number in the table does not include humanoid trajectories. Now we are working on RL-finetuning on humanoid platform, we will provide additional results using humanoid robots as soon as possible when they are ready. This combination enables more reliable transfer to the humanoid, code would be made publicly available. And we will add the analysis in the main paper with latest results as well.
>
> > ## Q4: The layout realism of modified URBAN-SIM environments.
>
> To address the concern regarding layout realism, we provide a quantitative analysis. We extract real-world layout statistics (e.g., longitudinal distributions) by applying Master3R, SAM2, and CLIP with carefully designed category words to 100 YouTube city-walking videos. This allows us to compute the longitudinal distribution of obstacles along the road centerline.
>
> ### Object Spacing Statistics
> | Category        | Mean (m) | Std (m) | Min (m) | Max (m) |
> |-----------------|------------------------|---------|---------|---------|
> | Tree            |       7.73                 |    3.61     |   3.11      |     20.0 (clipped)    |
> | Lamp            |       10.25                |    6.27    |      4.51    |      20.0 (clipped)   |
> | Trash bin       |     9.33  |    7.36     |  0.31     |     20.0 (clipped)   |
> | Traffic cone |          6.67              |   4.39      |    0.44     |    20.0 (clipped)     |
> | Advertising board |          8.98             |   7.25     |    0.05    |    20.0 (clipped)     |
>
> [Longitudinal distribution refers to the distance interval along the walking direction of the sidewalk or road (i.e., in the direction of robot motion).]
>
> (It's notable that lane and sidewalk segments do not have a fixed length; they are determined by the duration of the original video. In simulator, we typically set them to 30–50 m segments, which correspond to about 30–50 s for a robot moving at 1 m/s, and the width of sidewalk is sampled from unifrom(1.5m, 4.5m))
>
> We implement a procedural generation pipeline and tune its hyperparameters according to the extracted statistics. The code will be made publicly available. But it's notable that scene generation itself is not the focus of our work; it provides a means to create diverse training scenarios.

---

> > ### Author Response · Authors · 2025-11-24
> > **Response to Reviewer FY2s - Part 2**
> >
> > ## Questions
> > > ## Q5: Plan of 3D integration.
> >
> > Thanks for the insight. Incorporating 3D perception is a natural next step. Under the RGB-only setting, our plan is to first integrate metric-scale depth using a pretrained backbone (e.g., DepthAnything3 or Pi3) to obtain geometric information; then distill these geometric priors into a lightweight RGB encoder to reduce model size; freeze the encoder during policy training for stability; and finally deploy the model in real time using ONNX-based compression.
> >
> > > ## Q6: Expansion of benchmark scenarios.
> >
> > Yes, we plan to expand NavBench-GS with scenarios that better capture real urban layouts and dynamics. Concretely, we will (1) extend the benchmark with more narrow, obstacle-dense sidewalk environments, which more closely match the deployment settings of mobile robots than lane-based scenes, and (2) select new scenarios using measurable criteria such as interaction density, obstacle diversity, and spatial coverage. Since robust 4D reconstruction remains challenging compared to 3D, we will omit full 4D reconstruction in these new scenarios but record human trajectories and instantiate controllable pedestrian assets to simulate such dynamic behaviors, based on current compositional rendering and simulation pipeline.

---

### Official Review · Reviewer_vgu6 · 2025-11-01

**Soundness:** 3
**Presentation:** 3
**Contribution:** 3
**Rating:** 8
**Confidence:** 3

**Summary:**

This paper presents results from RL-finetuning of a navigation policy that is pretrained by SFT over static datasets. It empirically shows the improvements afforded by RL, introduing a way to provide data diversity and an architecture for the policy.

**Strengths:**

An unusual combination of SFT and RL for learning-based navigation - while existing approaches focus on one or the other, this paper shows the value of doing both.

**Weaknesses:**

The anchor description is not very clear. It seems to be constant-curvature arcs - a clearer explanation is needed.
Unclear how the constant curvature arcs approximate the full diversity of demonstration paths - is the matching performed instantaneously (i.e., for an instantaneous vx, w command mapped to the corresponding curvature?)

**Questions:**

Can you explain how the anchors are defined and chosen?

---

> ### Author Response · Authors · 2025-11-24
> **Response to Reviewer vgu6**
>
> We thank the reviewer for the constructive and insightful feedback. We are encouraged that the reviewer acknowledged the contribution of our approach. We address the concerns as follows.
>
> ## Weaknesses and Questions
> > ## Q1: Details of anchor design.
>
> We thank the reviewer for pointing out the need for a clearer anchor description.  Briefly, we generate M=64 representative anchor points by running **K-Means** over the ending points of all training trajectories in the unified dataset. These anchors (shown in **Figure 16** at **Page 22** of the revised paper) form a low-dimensional and structured representation of the underlying trajectory manifold, indicating the context-free distribution of the robot behaviors. A neural network $F_{\theta}$ then learns to map these anchors $p_a$ to GMM parameters based on the condition of RGB observations and optional goal information, which can be viewed as a **refinement** process that adjusts anchor points and the selects most appropriate mode to produce the final trajectory. Specifically,
>
> ### **Intuition**
> - Anchors provide a low-dimensional **structured basis** extracted from real trajectories.
> - The GMM enables **soft selection**, and is friendly for optimization.
> - The neural network is used for **refinement**, capturing diverse motion patterns, not limited to constant curvature arcs.
> - The mapping is **not an instantaneous curvature lookup**, but a probabilistic refinement conditioned on visual observations and goal.
>
> ### **Details of anchor generation and usage**
>
> | Stage | Description |
> |-------|-------------|
> | **1. Dataset collection** | Extract all 2D intention points from training trajectories. These are ground-truth waypoints in the robot frame. |
> | **2. K-Means clustering** | Run K-Means on all collected intention points to obtain **M** representative anchors. |
> | **3. Pre-generate anchor set** | The final anchor set is fixed and visualized in **Figure 16**. |
> | **4. Training** | Given observations $o_{t-k+1:t}$, goal information $p_d$ [optional], anchors $p_a$. Model outputs a distribution $GMM=F_{\theta}(p_a;o_{t-k+1:t},p_d)$, parameters of GMM are used for training as given in **Page 5**. |
> | **4. Inference** | Given observations $o_{t-k+1:t}$, goal information $p_d$ [optional], anchors $p_a$. Model outputs a distribution $GMM=F_{\theta}(p_a;o_{t-k+1:t},p_d)$, where the robot behavior is sampled from. |
>
>
> Concretely, we use an **anchor encoder** $F_{EA,\theta}$ and a **trajectory decoder** $F_{DT,\theta}$.
> Each anchor is first mapped into a hidden representation:
>
> $$
> h_a = F_{EA,\theta}(p_a),
> $$
>
> and then decoded into a set of future trajectory candidates and their mixture probabilities:
>
> $$
> \{w, q \} = F_{DT,\theta}(h_a; o_{t-k+1:t}, p_d),
> $$
>
> where the decoder is conditioned on the recent observation window and (optionally) the goal. Here, $p_a\in\mathbb{R}^{M\times2}, w\in\mathbb{R}^{M\times T\times2}, p\in\mathbb{R}^{M\times1}$, and $T$ is the prediction horizon. The controller (either PD controller or MPC controller) would be used to generate robot velocity and angular velocity based on the trajectory sampled from $\{ w, q \}$. The full processing pipeline and controller implementation will be made publicly available.

---

### Official Review · Reviewer_SfZD · 2025-11-01

**Soundness:** 2
**Presentation:** 2
**Contribution:** 2
**Rating:** 4
**Confidence:** 3

**Summary:**

This paper proposes S2E (Seeing-to-Experiencing), a framework for training navigation foundation models that combines offline pretraining on real-world videos with reinforcement learning (RL) fine-tuning in simulation. The key technical contributions include: (1) Anchor-Guided Distribution Matching (AGDM) for modeling multimodal navigation behaviors during pretraining, (2) a Residual-Attention Module (RAM) that enables RL fine-tuning while preserving pretrained knowledge, and (3) NavBench-GS, a benchmark built on 3D Gaussian Splatting scenes. The authors claim that RL alleviates diminishing returns from scaling offline data alone and enables zero-shot transfer to real-world scenarios.

**Strengths:**

Comprehensive System: The paper presents an end-to-end system from data collection through real-world deployment, which requires significant engineering effort.

Cross-Embodiment Evaluation: Testing on wheeled, quadruped, and humanoid robots (Table 6) demonstrates some generality, though the humanoid results are quite poor.

Honest Limitations Discussion: Section 5 acknowledges the limitation of vision-only approaches and collision failures.

Detailed Implementation: The appendix provides extensive implementation details, which aids reproducibility.

**Weaknesses:**

1. Lack of Theoretical Insight: The paper doesn't explain why RL helps beyond showing empirical improvements. What specific failure modes of offline learning does RL address? What inductive biases make certain behaviors learnable only through interaction?

2. Modest Improvements: Many reported improvements are marginal (e.g., Table 2: 0.51 vs. 0.32 success rate for wheeled robot). Given the additional computational cost of RL (8 hours on L40S GPU), the cost-benefit tradeoff is unclear.
Cherry-Picked Comparisons:

3. CityWalker* (retrained on same 100h data) actually achieves 0.67 SR vs. S2E's 0.82 SR in empty scenes (Table 1), suggesting the RL contribution is partially due to other factors;
Figure 6(a) compares against "prior methods" shown as dotted lines, but these prior methods use different evaluation protocols


4. Limited Failure Analysis: The paper shows successful cases but doesn't systematically analyze failure modes. When does RL fine-tuning hurt performance? What environments or scenarios remain challenging?

5. Questionable Design Choices:

Why freeze the visual encoder during RL? This prevents learning better visual features for interaction
Why use a simplified entropy approximation (Eq. 11) instead of more accurate estimators?
The stochastic goal masking strategy (Section E.2) seems arbitrary - no ablation justifies the specific probabilities chosen


6. Reproducibility Concerns: Despite promising code release, the method depends on:
Proprietary Unitree robot APIs
Multiple datasets with different licensing terms
NVIDIA IsaacSim which requires expensive GPU resources
Hand-tuned reward functions that may not transfer to other environments

7. Missing Related Work

The paper should discuss and compare with:

Offline RL for Robotics: Kumar et al. (2020, NeurIPS), Nair et al. (2020, CoRL), Mandlekar et al. (2021, CoRL)
Vision-Language-Action Models: Driess et al. (2023, ICML - PaLM-E), Brohan et al. (2023, CoRL - RT-2)
Navigation Benchmarks: Savva et al. (2019, ICCV - Habitat), Xia et al. (2018, CVPR - Gibson)
Sim-to-Real Transfer: Peng et al. (2018, ICRA), Tan et al. (2018, CoRL)

**Questions:**

Data Efficiency: How many RL environment steps are needed? What's the sample complexity compared to collecting more offline data?

Reward Engineering: How sensitive is performance to reward function design? Have you tried learning rewards from human preferences (Christiano et al., 2017)?

Failure Modes: What percentage of real-world failures are due to:

Perception errors (misdetecting obstacles);
Planning failures (local minima);
Control errors (locomotion instability);
Sim-to-real gap


Comparison Fairness: Can you provide results where all methods are:

Trained on identical 100h dataset;
Evaluated on an established benchmark (not self-proposed);
Using the same evaluation protocol and metrics


Generalization: How does performance degrade with:

Different camera intrinsics/extrinsics;
Different weather/lighting conditions;
Novel obstacle types not seen in training


RL vs. SFT: The paper claims RL is better than SFT (Figure 6b), but:

How was the SFT data collected? From the pretrained policy or optimal demonstrations?
Did you try other offline RL algorithms (CQL, IQL, etc.) that might bridge the gap?

---

> ### Author Response · Authors · 2025-11-24
> **Response to Reviewer SfZD - Part 1**
>
> We thank the reviewer for the constructive and insightful feedback. We are encouraged that the reviewer acknowledged the practical value of our system ("significant engineering effort"), and the clarity and reproducibility of our work ("honest limitations discussion", "detailed implementation"). We address the concerns as follows.  (We provide additional visualization results **[Anonymous Video Link](https://h1h2h3h4h5h.github.io/)** to address the concerns from the reviewer.)
>
> ## Weaknesses
> > ## Q1: Theoretical insight.
>
> Thank you for the insightful and constructive comment. We agree that the paper benefits significantly from a deeper theoretical justification for the necessity of RL. We have revised **Section 3.2 (Page 5, 6)** to explicitly derive the failure modes of offline learning. Below, we provide the concrete theoretical details regarding the interplay between **Compounding Error** and **Off-Manifold States**.
>
> **1. Failure Mode: The fundamental limitation of offline training arises from the compounding errors**.
>
> In offline learning, the policy $\pi_{\theta}$ is trained to minimize prediction error given expert states $s_t^E \sim d_{\pi_E}$, which ignores the change in distribution. However, during deployment, small approximation errors $\epsilon$ in the policy accumulate over time. As established in (Ross et al., 2010, Ross et al., 2011), this can lead to a distribution drift that scales quadratically with the horizon $T$ in some scenarios:
>
> $$
> E_{s' \sim d_{\pi_\theta}} [Dist(s', d_{\pi_E})] \propto T^2 \epsilon.
> $$
>
> This compounding error can push the policy trajectory away from the training distribution $d_{\pi_E}$, where $Dist(\cdot)$ measures the divergence between the state distribution from the policy and the expert distribution. As the deviation grows, the policy encounters states on which no supervision was provided, causing further drift and making closed-loop behavior degraded.
>
> Since expert demonstrations inherently lie on a low-dimensional manifold of "perfect" behavior, $d_{\pi_E} = \{ (s_t^E, a_t^E) \}$. Crucially, the drifted states caused by compounding errors such as near-collision, misalignment are absent from this manifold. Let $S_{\text{off-manifold}}$ denote the set of these drifted states. In the offline dataset, the probability mass on these states is near zero:
>
> $$p(s\in S_{\text{off-manifold}}\mid d_{\pi_E}) \approx 0.$$
>
> Consequently, supervised learning provides **no supervision** for corrective behaviors. The policy $\pi_{\theta}$ from offline training is undefined for critical safety skills such as:
> * Obstacle avoidance after drifting off-path.
> * Re-centering on the sidewalk after a control glitch.
>
> This theoretical gap explains why supervised models fail: **compounding errors** drive the agent into $S_{\text{off-manifold}}$, and **mainfold mismatch** ensures the agent has no learned gradient to recover from those states.
>
> RL explicitly solves this by shifting the optimization objective to the **induced distribution** $d_{\pi}$.
>
> $$
> \max_\pi E_{s \sim d_{\pi}} [ Q^{\pi}(s, a) ].
> $$
>
> Unlike the static $d_{\pi_E}$, the induced distribution $d_{\pi}$ naturally covers $S_{\text{off-manifold}}$ due to exploration. This forces the value function to learn valid estimates in perturbed regions, providing the necessary gradient signal to correct the compounding errors and guide the agent back to the safe manifold.

---

> > ### Author Response · Authors · 2025-11-24
> > **Response to Reviewer SfZD - Part 2**
> >
> > > ## Q2: Modest improvements.
> >
> > Thank you for raising this point. While some of the absolute gains may appear modest, even small improvements often translate into substantial reductions in failure cases such as collisions. We have added more qualitative visualizations in the **[Anonymous Video Link](https://h1h2h3h4h5h.github.io/)**, demonstrating that our method is noticeably robust in complex environments.
> >
> > And we now include the ablation study covering SFT, DecFT-RL, FullFT-RL, PPO, and our method. Here, we provide the expanded results of Table 3 (on Navbench-GS-Obstacle), Reinforcement learning with full-parameter fine-tuning (FullFT-RL).
> >
> > | Methods | SR $\uparrow$  | CT $\downarrow$ |
> > | :--- | :---: | :---: |
> > | PPO | 0.02 | 2.37 |
> > | SFT | 0.49 | 0.77 |
> > | DecFT-RL | 0.39 | 0.91 |
> > | FullFT-RL | 0.31 | 1.33 |
> > | **Ours** | **0.57** | **0.69** |
> >
> > All models are evaluated under identical conditions, and the results consistently show that our approach outperforms the baselines across diverse scenarios.
> >
> >
> > > ## Q3: Detailed failure mode analysis.
> >
> > Thank you for providing this insightful comment. We added more qualitative visualizations in the **[Anonymous Video Link](https://h1h2h3h4h5h.github.io/)** to analyze the failure modes of our approach.
> >
> > Since our policy is trained end-to-end from RGB observations, it is inherently difficult to isolate perception errors from planning errors, as they are jointly represented in the network. Nevertheless, we performed an additional failure categorization analysis on the real-world evaluations to provide more clarity.
> >
> > We divide failures into two actionable categories:
> >
> > - Control failures, where the robot executes unstable or overly aggressive low-level motions (e.g., oscillations or over-steering);
> >
> > - Model failures, where the policy generates incorrect high-level decisions (e.g., drifting toward non-walkable regions or failing to react to a pedestrian).
> >
> > For road-structure understanding (e.g., curved paths, narrow sidewalks), we observe that 1 out of 10 failures are due to model errors, while 9 out of 10 arise from controller instability. These typically manifest as over-steering at tight turns or oscillations when correcting lateral deviation.
> >
> > For obstacle-avoidance, the failure distribution is approximately 5 out of 10 failures stem from high-level model mistakes (e.g., misjudging obstacle passability or choosing a suboptimal avoidance direction), whereas the remaining 5 out of 10 come from the low-level controller (e.g., insufficient braking or turning radius limitations).
> >
> > Because our policy is trained end-to-end from RGB observations, perception and planning errors are inherently entangled within the “model failure’’ category, making fully disentangled attribution difficult.

---

> ### Author Response · Authors · 2025-11-24
> **Response to Reviewer SfZD - Part 3**
>
> > ## Q4: Why freeze the visual encoder during RL?
>
> Thank you for this insightful question. We freeze the visual encoder during RL because updating it with simulated RGB observations can easily cause feature domain drift and degrade the pretrained visual representations, which are critical for generalization. As discussed in the **Section 3.2** in the revised paper, the simulator’s texture, lighting, and structure differ significantly from real-world data; fine-tuning the encoder solely on simulated images leads to overfitting to the simulation domain and harms downstream deployment performance. By freezing the encoder, the RL stage focuses on learning interaction-related corrections while preserving the robust visual features learned from large-scale real data.
>
> We include additional quantitative results in the extended Table 3, comparing frozen-encoder RL (DecFT-RL, our method) with full-parameter RL (FullFT-RL). These results show that updating the encoder during RL noticeably reduces success rate and increases collisions, further supporting our conclusion.
>
> > ## Q5: Simplified entropy approximation (Eq. 11)
>
> Thank you for pointing this out. We clarify the motivation for our entropy approximation here.
>
> For a **Gaussian Mixture Model** (GMM) over actions $w$
>
> $$\boldsymbol{q}(w) = \sum_{m=1}^{M} q_m, \mathcal{N}(w \mid \mu_m, \Sigma_m),$$
>
> the exact differential entropy is
>
> $$H(\boldsymbol{q}) = - \int \boldsymbol{q}(w) \log \boldsymbol{q}(w) dw.$$
>
> However, unlike a single Gaussian, there is no known closed-form analytical solution for the differential entropy of a GMM. The difficulty arises from the definition of differential entropy: for a GMM, the probability density function is a weighted sum of Gaussians, when try to substitute this into the entropy formula, the term $\log (\sum_{m=1}^M\boldsymbol{q}_m\mathcal{N}_m(w))$ (the **log of a sum**), unlike the log of a product, the log of a sum cannot be expanded into simpler terms, making the integral intractable to solve analytically. Therefore, computing ($H(\boldsymbol{q})$) exactly would require numerical integration or Monte Carlo estimation, which is expensive and noisy during RL training.
>
> To keep the objective stable and efficient, we adopt an approximation that assumes limited overlap between mixture components. In that case the mixture entropy can be approximated by the entropy of the categorical weights plus the weighted entropies of each Gaussian given in our Eq. (11) follows this approximation. The goal is not to obtain an exact entropy, but to provide a **tractable and smooth regularizer** that encourages sufficient exploration over modes without introducing significant computational overhead.
>
> ## Q6: Stochastic goal masking strategy
>
> Thank you for asking about the motivation behind our stochastic goal masking.
>
> In real-world experiments, the robot receives a **very noisy GPS-like goal**, which can have errors larger than 5 m on our platforms. However, during supervised training, we have access to **exact ground-truth goals** ($p_d$).
> In many collected data, this goal is almost a deterministic function of the future robot pose $x_{t+\Delta t}$:
> $$
> p_d \approx x_{t+\Delta t},
> $$
> and the input to the policy is often the **relative goal** in the robot frame. As a result, there is an issue where the model can largely ignore the visual observation $o_t$ and instead learn a mapping
> $$
> w_t \approx f(p_d),
> $$
> which is highly predictive under clean goals. This leads to a **shortcut learning** issue: the policy overrelies on the goal input, which is clean during training but highly noisy at deployment, and therefore does not learn to use the scene context from the observation $o_t$. To mitigate this, we use the **stochastic goal masking strategy**. With probability $p_{\text{mask}}$, the goal information is removed (or heavily down-weighted), forcing the model to rely on visual information. With probability $1 - p_{\text{mask}}$, the clean goal is available, so the model still learns to exploit goal information when it is reliable.
>
> We provide additional experimental results to illustrate the effectiveness of the strategy.
>
> | Strategy | Success / Total |
> |-------|-------------|
> | **Goal w/o masking** | 3 / 10 |
> | **Goal w masking** | 8 / 10 |
>
> This experiment is conducted on the scenario shown in the long horizon navigation in supplementary video. Under noisy GPS conditions where the goal is occasionally misplaced onto non-walkable regions such as grass, the model trained without masking often fails to complete the turn. In contrast, the model trained with masking relies more on visual information, enabling it to follow the walkable path and successfully complete the task despite the incorrect goal input.

---

> ### Author Response · Authors · 2025-11-24
> **Response to Reviewer SfZD - Part 4**
>
> > ## Q7: Reproducibility concerns.
>
> Thank you for the constructive suggestion. We address the different aspects of reproducibility concerns separately below.
>
> #### (1)  Unitree robot APIs
>
> Our method is not dependent on any proprietary Unitree-specific API logic. We only utilize the standard low-level interface for sending joint commands during real-world deployment, which is not required to reproduce our RL training or Navbench-GS benchmark results. To ensure full reproducibility, we will release all code for training, inference, and evaluation, including the neural network-based locomotion controller used in our experiments. Since the model outputs trajectories, the policy can be deployed on any robot with a standard velocity/steering interface. At the same time, the controller we used in real-world deployment would be made publicly available as well.
>
> #### (2) Multiple datasets with different licensing terms
> All datasets used in our experiments are fully open-sourced and available for academic research. There are no licensing restrictions that prevent reproduction of our results, and all preprocessing scripts will be released to ensure the training data can be reconstructed.
>
> #### (3) NVIDIA IsaacSim requirement
> While IsaacSim leverages GPU acceleration, it does not strictly require enterprise-grade hardware. Our model can be used effectively on standard consumer-grade GPUs (e.g., a single RTX 4090 or 5090). To ensure broad accessibility and reproducibility, we will release the pre-generated datasets collected from Urban-Sim to address the issue.
>
> #### (4) Hand-tuned reward functions
> Our framework is not reward-sensitive, the reward function can be very generic, consisting primarily of a success reward and a collision penalty. To address concerns regarding tuning, we provide ablation studies here. While we utilize a mild shaping term in the paper, our method is capable of learning effectively with only sparse signals (i.e., strictly success and collision).
>
> | Strategy | SR | CT |
> |-------|-------------|-------------|
> | **Full** | 0.57 | 0.69 |
> | **Sparse** | 0.51 | 0.62  |
>
> The evaluations are conducted on NavBench-GS-Obstacle. The Full setting corresponds to the reward used in the main paper, while Sparse only used success reward and collision penalty. The results demonstrate that our method maintains stable performance even without shaping, indicating that it does not highly rely on reward engineering. We agree that robustness to reward design is an important aspect, and we will do more discussion regarding this.
>
> > ## Q8: Comparisons with more related works.
>
> Thank you for the constructive comment. We agree that including a broader set of related works can strengthen the paper. We are working on and will incorporate comparisons with additional offline RL methods and other closely related approaches in learning-based navigation. The updated results will be included as soon as they are ready.
>
> ## Questions
> > ## Q9: Data efficiency.
>
> Thank you for the question. Our RL fine-tuning phase uses approximately $1\times10^5$ environment steps, as shown in **Figure 15** in the Appendix. These samples can be generated efficiently in Urban-Sim, which runs at hundreds of FPS across multiple parallel environments.
>
> In contrast, collecting additional real-world offline data is significantly more costly. More importantly, certain states that are critical for robust navigation such as near-collision, off-center drift, or other failure-recovery scenarios are unsafe to collect in the real world. Offline datasets therefore underrepresent these rare but important states, making it difficult for purely supervised training to learn recovery behaviors.
>
> Simulator can generate these safety-critical states automatically and improving policy robustness without requiring additional risky or expensive real-world data collection.
>
> > ## Q10: Reward engineering.
>
> Thank you for the constructive question. We provide additional analyses and ablations regarding the reward design in **Q7(4)** of the response.
>
> > ## Q11: Comparison fairness.
>
> Thank you for the constructive comment. We agree that fair comparison is important, and we are currently running additional evaluations on well-established benchmark systems. These results will be updated as soon as they become available.

---

> ### Author Response · Authors · 2025-11-24
> **Response to Reviewer SfZD - Part 5**
>
> > ## Q12: Generalization issues.
>
> Thank you for raising this important point. We address each aspect of generalization separately below.
>
> #### (1) Camera configuration.
> The Unitree Go2 dataset was never used during any training stage (neither supervised training nor RL fine-tuning). The model is deployed directly on Go2 with the same weights, without camera-specific tuning. We acknowledge that camera differences (FOV, mounting height, lens distortion) may affect performance in certain cases. Following your suggestion, we plan to simulate a broader range of camera intrinsics in future versions to further improve robustness.
>
> #### (2) Weather and lighting variations.
> We provide additional visualizations in the updated **[Anonymous Video Link](https://h1h2h3h4h5h.github.io/)**, showing the model behavior under diverse lighting conditions, including shadows, low-light regions, and sunlight variations in urban scenes. Although these conditions were not explicitly augmented during training, the pretrained visual encoder already provides reasonable robustness.
>
> #### (3) Novel obstacle types.
> As shown in the **[Anonymous Video Link](https://h1h2h3h4h5h.github.io/)** (1.2), certain objects such as vehicles are not included in the simulated scenarios. Nevertheless, our model still shows reasonable avoidance behaviors when encountering previously unseen obstacles. This aligns with our goal of training a policy that generalizes across scene variations without requiring obstacle-specific training.
>
> We will clarify these points in the revised paper and continue expanding our work to include more diverse environments and sensor configurations.
>
> > ## Q13: Details of SFT.
>
> Thank you for the insightful question. We provide the detailed process of SFT experiments in our paper here.
>
> ### **Details of data collection process in Urban-Sim for SFT**
>
> | Stage                        | Description |
> | ---------------------------- | ---------------------------------------------------------------------------------------------------------------------------------------------------------------------------------------------------------------- |
> | **1. Rollout generation**    | We use the pretrained model to navigate in all Urban-Sim training scenarios. At each step, the model outputs an action distribution, and the executed action is sampled from this distribution.|
> | **2. Closed-loop execution** | The robot follows the sampled actions to generate full navigation trajectories in a closed-loop manner. This produces realistic on-policy behaviors that reflect the pretrained decision-making pattern of model.|
> | **3. Success filtering**     | Only trajectories that successfully reach the goal without collisions are kept. Failed rollouts (e.g., stuck, drifting, or colliding cases) are discarded to ensure clean and reliable supervision. |
> | **4. Dataset construction**  | The remaining successful trajectories are aggregated into the SFT dataset, consisting of observation–action pairs of the form ((o_t, a_t)). These serve as expert demonstrations for supervised fine-tuning.|
> | **5. SFT training**          | SFT minimizes a per-step supervised loss over this dataset, producing an imitation-based policy initialized from on-policy rollouts of the pretrained model.|
>
> And the visualization results of anchor point distribution is now provided in the **Figure 16** in the Appendix.

---

### Author Response · Authors · 2025-12-02
**Overall Summary (2/2)**

---

**Key Concerns and How We Addressed Them:** Alongside the positive feedback on **S2E**, reviewers raised several questions regarding additional ablation studies, theoretical insight and clearer description. We addressed each point in detailed individual responses and updated the manuscript accordingly.  Below, we concisely summarize the main concerns and how we addressed them:

---

> **Q1:** Can S2E work under different/extrerm weather/lightning conditions, more complex traffic conditions with novel obstacle types? (asked by `SfZD` / `FY2s`)

Yes. And to address these concerns, we provide additional real-world closed-loop evaluation videos in the [Anonymous Video Link](https://h1h2h3h4h5h.github.io/). Our model demonstrates robust avoidance behaviors when encountering previously unseen obstacles as well as extreme weather and lighting variations—including shadows, low-light regions, glare, and strong sunlight. These results support our claim that S2E learns a policy capable of generalizing across diverse scene conditions without requiring obstacle-specific fine-tuning.

> **Q2:** Details of comparison between SFT and RL. (asked by `SfZD` / `UWpN`)

We provide additional details on the complete SFT pipeline in the [response](https://openreview.net/forum?id=0c7nAZjyr5&noteId=iDi0cKmeT6), as well as a more formal theoretical analysis in the [response](https://openreview.net/forum?id=0c7nAZjyr5&noteId=KJ3GMojw4A). The expanded explanation clarifies why SFT alone still suffers from overfitting to the expert data distribution and is vulnerable to covariate shift during deployment. In contrast, RL explicitly optimizes closed-loop performance, enabling the policy to correct for distributional shift, recover from compounding errors, and improve decision-making robustness.

> **Q3:** Clarification of key innovations. (asked by `SfZD` / `vgu6` / `UWpN`)

We have substantially revised the content on pages 5–6 of the updated paper to more clearly motivate the key innovations of S2E and to provide a more formal analysis of the core component, i.e., residual attention module.

More details are provided in the rebuttal and the revised paper, where we have highlighted all changes for clarity.

We again thank the Area Chairs and Reviewers for their time and effort. Finally, we sincerely hope the Area Chairs will consider S2E’s contributions to the community, along with the detailed rebuttal responses and the improved revised manuscript that addresses all reviewer concerns.

Best regards,

Authors of Submission #9621

---

### Author Response · Authors · 2025-12-02
**Overall Summary (1/2)**

We sincerely thank  Area Chairs and the Reviewers for their time, careful evaluations, and constructive feedback. Their comments helped us substantially strengthen the revised version of **S2E**, now posted. We especially appreciate the tremendous efforts of the newly assigned Area Chairs in carefully reviewing our rebuttal and revised manuscript.

In the following overall summary, we provide a concise overview of the paper’s core contributions, summarizes the reviewers’ feedback and main concerns, and highlights how each point was addressed and incorporated into the revised manuscript.

---

## ***Paper and Contribution Summary***

---

In summary, we introduce the Seeing-to-Experiencing (S2E) learning framework to scale the capability of navigation foundation models with reinforcement learning. **S2E** combines the strengths of pre-training on offline videos and post-training through reinforcement learning. It maintains the model's generalizability acquired from large-scale real-world videos while enhancing its interactivity through reinforcement learning in simulation environments. Specifically, we introduce two innovations:

1. an Anchor-Guided Distribution Matching strategy for offline pretraining, which stabilizes learning and models diverse motion patterns through anchor-based supervision;
2. a Residual-Attention Module (**RAM**) for reinforcement learning, which obtains reactive behaviors from simulation environments without erasing the model’s pretrained knowledge.

Moreover, we establish a comprehensive end-to-end evaluation benchmark, **NavBench-GS**, built on photorealistic 3D Gaussian Splatting reconstructions of real-world scenes that incorporate physical interactions. It can systematically assess the generalizability and safety of navigation foundation models.

---

## ***Review Summary, Updates and Improvements from the Rebuttal***

---

**Initial Reviews and Rebuttal Exchanges**: **Our work, S2E, received *initial* ratings of 8 (`vgu6`), 6 (`UWpN`), 4 (`SfZD`), and 4 (`FY2s`).** After we posted our detailed rebuttal responses, reviewer `UWpN` [replied](https://openreview.net/forum?id=0c7nAZjyr5&noteId=HyB5iDVcLE) and *reaffirmed his/her score of* ***6***. The remaining reviewers did not have time to respond our rebuttal before the reviewer-response window closed.

---

**Reviewer Consensus on Key Strengths:** From the initial reviews, we are pleased that the core contributions of **S2E** were recognized. Reviewer `SfZD` highlighted the **comprehensive** end-to-end system, the cross-embodiment evaluations, the **honest** discussion of limitations, and the **detailed implementation** that improves reproducibility. Reviewer `FY2s` emphasized our three main contributions: the Anchor-Guided Distribution Matching (AGDM) for modeling multimodal navigation trajectories, the Residual-Attention Module (RAM) for RL fine-tuning without losing pretrained capabilities, and the NavBench-GS benchmark enabling realistic closed-loop evaluation, **solving** the reproducibility issue of real-world testing. Reviewer `vgu6` acknowledged that we introduce a way to provide data diversity and an architecture for the policy. Reviewer `UWpN` recognized the crucial role of RL (**extremely helpful**) in enhancing closed-loop navigation, the importance of anchor-based GMM action parameterization with in-depth analysis, the empirical insights on data saturation, and the **superior** zero-shot real-world performance across different robot embodiments.

---

### Meta-Review · Area_Chair_xKJd · 2025-12-22

**Summary:**

The paper received divergent ratings (8,6,4,4). The reviewers raised various concerns such as (1) Lack of theoretical insights, (2) Modest improvements, (3) Lack of clear descriptions for anchors, (4) Relying only on visual data, (5) Limited real-world evaluations, (6) Lack of comparisons with diffusion models.

The rebuttal addressed the concerns well. For instance, it provided theoretical insights about limitations of offline training and results of diffusion baselines. Only Reviewer UWpN responded before the discussion cut-off and confirmed their 6 rating. Overall, the AC appreciates the two main contributions of the work, namely, AGDM to make the learning of multi-modal distributions more robust, and the residual component that incorporates information from interactions. Both contributions are interesting and will be beneficial for the research community. Hence, acceptance is recommended.

**Reviewer Concerns:**

Refer to the box above.

**Reviewer Scores:**

Reviewer UWpN checked the rebuttal and confirmed their 6 rating.

The concerns regarding relying only on visual data, real-world evaluations, lower success of the humanoid robot, and details of URBAN-SIM modifications have been addressed well. Reviewer FY2s could increase the rating.

 Reviewer vgu6 needed clarification regarding anchor description. It is addressed. So, they would probably keep the rating.

The concerns regarding lack of theoretical insights, modest improvements, failure mode analysis, design choices, reproducibility and missing related work are convincingly addressed. Reviewer SfZD could potentially improve the rating.

---

### Decision · Program_Chairs · 2026-01-26

Accept (Poster)